# Braking performance oriented multi–objective optimal design of electro–mechanical brake parameters

**Tong Wu** [1], **Jing Li** [2]*, **Xuan Qin** [3]

1 College of Automotive Engineering, Jilin University, Changchun, Jilin, China, 2 School of Vehicle and Energy, Yanshan University, Qinhuangdao, Hebei, China, 3 School of Automotive Engineering, Hubei University of Automotive Technology, Shiyan, Hubei, China

☯ These authors contributed equally to this work.

* l_jing1129@163.com

**Data Availability Statement:** All relevant data are within the manuscript and its Supporting Information files.

**Funding:** JL received support from the National Key R&D Program of China (2018YFB0105900).

## Abstract

Excellent braking performance is the premise of safe driving, and improve the braking performance by upgrading structures and optimizing parameters of braking systems has become the pursuit of engineers. With the development of autonomous driving and intelligent connected vehicle, new structural schemes such as electro–mechanical brakes (EMBs) have become the future of vehicle braking systems. Meanwhile, many scholars have dedicated to the research on the parameters optimization of braking systems. While, most of the studies focus on reducing the brake size and weight, improving the brake responses by optimizing the parameters, almost not involving the braking performance, and the optimization variables are relatively single. On these foundations, a multi–objective optimal design of EMB parameters is proposed to enhance the vehicle's braking performance. Its objectives and constraints were defined based on relevant standards and regulations. Subsequently, the decision variables were set, and optimal math model was established. Furthermore, the co–simulation platform was constructed, and the optimal design and simulation analyses factoring in the crucial structural and control parameters were performed. The results confirmed that the maximum braking pressure response time of the EMB is decreased by approximately 0.3 s, the stopping distance (SD) of 90 km/h–0 is shortened by about 3.44 m. Moreover, the mean fully developed deceleration (MFDD) is increased by 0.002 g, and the lateral displacement of the body (LD) is reduced by about 0.037 m. Hence, the vehicle braking performance is improved.

## 1. Introduction

With the economic and social development, there are some trends such as flexible mechanical layouts, compatible control architectures and integrated functions in vehicle subsystems. Meanwhile, the automotive industries have also entered intelligent and connected frontiers [1, 2]. Thus, safety is the premise of autonomous driving and intelligent connected vehicle, and

**Competing interests:** The authors have declared that no competing interests exist.

constantly improving the safety also has become the eternal pursuit of automotive engineers. As an important standard to measure vehicle safety, the braking performance has gained increased attention in the public, and there are many ways to improve the braking performance.

In structural respects, electro–mechanical brakes (EMBs) with electric control and motor driving will become the future of vehicle braking. The EMBs hinder the brake pipelines and disconnects the mechanical connections between the brake pedals and the actuators completely, and the actuator operations at each wheel are driven by electric signals directly. Moreover, the EMBs demonstrate a convenience for integrating the autonomous and the advanced driving assistance systems to realize multiple braking modes. Particularly in commercial vehicles, the EMB does not require complicated pneumatic components, which could effectively reduce the body weight, eliminate the exhaust noise and realize more positive response, accurate control and distribution of the braking pressure [3]. It is understood that the braking performance of a bus equipped with an EMB could be improved by approximately 24.7% than a pneumatic brake [4].

Furthermore, the parameters optimization also an effective way to improve the braking performance. Several scholars have been investigating the parameters optimizations of brakes: Durali, Khajepour *et al.* proposed a cam–based brake, and the pressure response time was set as an objective to optimize the brake structural parameters [5]. Zhou, Gao *et al.* established a multi–objective optimal model for the braking time, the thicknesses and the temperature rises, and obtained the Pareto sets ultimately [6]. Anselma and Belingardi took the size and the energy recovery rate as the objectives, and defined the safety requirements as the constraints, and then optimized the structural parameters of a hydraulic brake [7]. In the Shamieh and Sedaghati's study, the dynamic range, the response time and the weight of a damper were taken as the objectives, and the size and the flux density were set as the constraints, and then the optimization was solved [8]. Wang, Zhao *et al.* optimized an electro–hydraulic brake parameters based on the objectives of the brake feeling and the energy recovery rate [9]. Khafaji and Manning tested the influence of the drum brake parameters and optimized the parameters based on the Taguchi method [10]. Zhang, Lu *et al.* introduced a differential evolution cellular multi–objective algorithm, and optimized the structural parameters of a drum brake aimed to improve the efficiency and reduce the volume and the temperature of the brake [11]. Yao, Miao *et al.* designed an electromagnetic wedge brake, and defined the maximum braking torque as an objective, and set the motor power and the braking feeling as the constraints, and optimized the wedge angle and the reduction ratio of the brake [12]. Considering the electromagnetic, the non–Newtonian flow characteristics and the heat transfer characteristics of the magnetorheological fluid, Topcu *et al.* proposed an improved multi–physical optimization of the rotary magnetorheological fluid brake [13]. In the [14], a neural network model of a composite braking system was established to improve the energy economy and the braking stability of an electric vehicle by optimizing the range of the regenerative and hydraulic brakes. In He and Wang's study, a dynamic model and braking force distribution strategy of the new brake were established, and a multidisciplinary collaborative optimization was proposed, and the control parameters were optimized [15]. In the [16], a regenerative braking control strategy was proposed, and the energy recovery distribution coefficient was optimized under the constraints of the regenerative braking power and the ideal braking distribution curve (I–curve). Liu, Lei *et al.* also proposed a regenerative braking control strategy, while the braking performance, the regenerative braking and battery loss rates were set as objectives [17]. Xu, He *et al.* conducted the theoretical analyses on energy recovery of a regenerative braking system, and analyzed the influence of different combinations in the braking strength, the initial pressure of accumulator and the initial velocity on energy recovery rate [18].

From the analyses, the optimal objectives in the [5–13] are the structural parameters, and the objectives in the [14–18] are the control parameters. Except for the [14] and the [17], the objectives in other studies are limited to improving the brake responses and reducing its size and weight.

Actually, the brake responses are influenced by both the structural and the control parameters. Furthermore, the vehicle's braking performance is not entirely determined by the brake responses, but also impacted by such factors as the load transfers, the braking distributions and the control effects. Hence, the ultimate goal of the optimization should be improve the vehicle's braking performance and not just the brake responses.

Therefore, it is important to perform a braking–performance oriented optimal design that considering both the structural and the control parameters of a braking systems. To fill the research gaps, an upgraded EMB scheme was proposed as a continuation of our previous study [3], and its system model and control prototype were built in this study. Subsequently, to improve the vehicle's braking performance, the objectives and constraints were defined based on relevant standards and regulations, and the decision variables were set from the structural and the control parameters. Finally, an optimal math model and a co–simulation platform were established, and the optimal design and simulation analyses were performed based on the non–dominated sorting genetic algorithm with elitism mechanism (NSGA–II).

Compared with other studies, the main contribution of this study is providing a new idea for the parameters optimization of vehicle braking systems. Specifically, the first is to consider both the structural and the control parameters when selecting the decision variables; the second is to set the optimal objectives from the perspective of vehicle braking performance, which is no longer limited to the brake responses. In these ways, the influences of the load transfers, the braking distribution and the parameters coupling can be decreased.

This paper is organized as follows: an upgraded EMB scheme and its working principle are introduced in Section 2; the optimal design and simulation analyses are performed in Section 3; the optimal results are analyzed and discussed in Section 4; conclusions are made in Section 5.

## 2. EMB structure, system model and control prototype

This section firstly introduces the structure and working principle of the improved EMB, then establishes the EMB system model and control prototype, and finally gives the parameters of the model and the prototype.

### 2.1 EMB structure and working principle

In our previous work, an EMB structural scheme based on pneumatic–disc–brake for a M3 electric bus has been proposed. The EMB is generally composed of a powerplant and a caliper. In this study, its structure was improved, as shown in Fig 1.

Fig 1A shows the structure of a pneumatic–disc–brake, and Fig 1B shows the improved EMB. The power transmission path of the pneumatic–disc–brake is shown with red arrows in Fig 1A. During the working of the brake, the pressured air is injected into the chamber through the inlet (10), and the diaphragm (9) and the push rod (6) are pushed by the air. At the same time, the spring (8) is compressed, and the push rod and diaphragm jointly change the air pressure into axial thrust and act on the arm (5) in the caliper. The lower end of the arm exerts an amplified pressure on the piston (4) and drives the piston to push the brake shoes (3) to press the disc (2) for generating the braking pressure.

Noted that the axial thrust output from the push rod (6) is amplified and so the size of the air chamber could be reduced due to the arm (5). Furthermore, the caliper already existed a

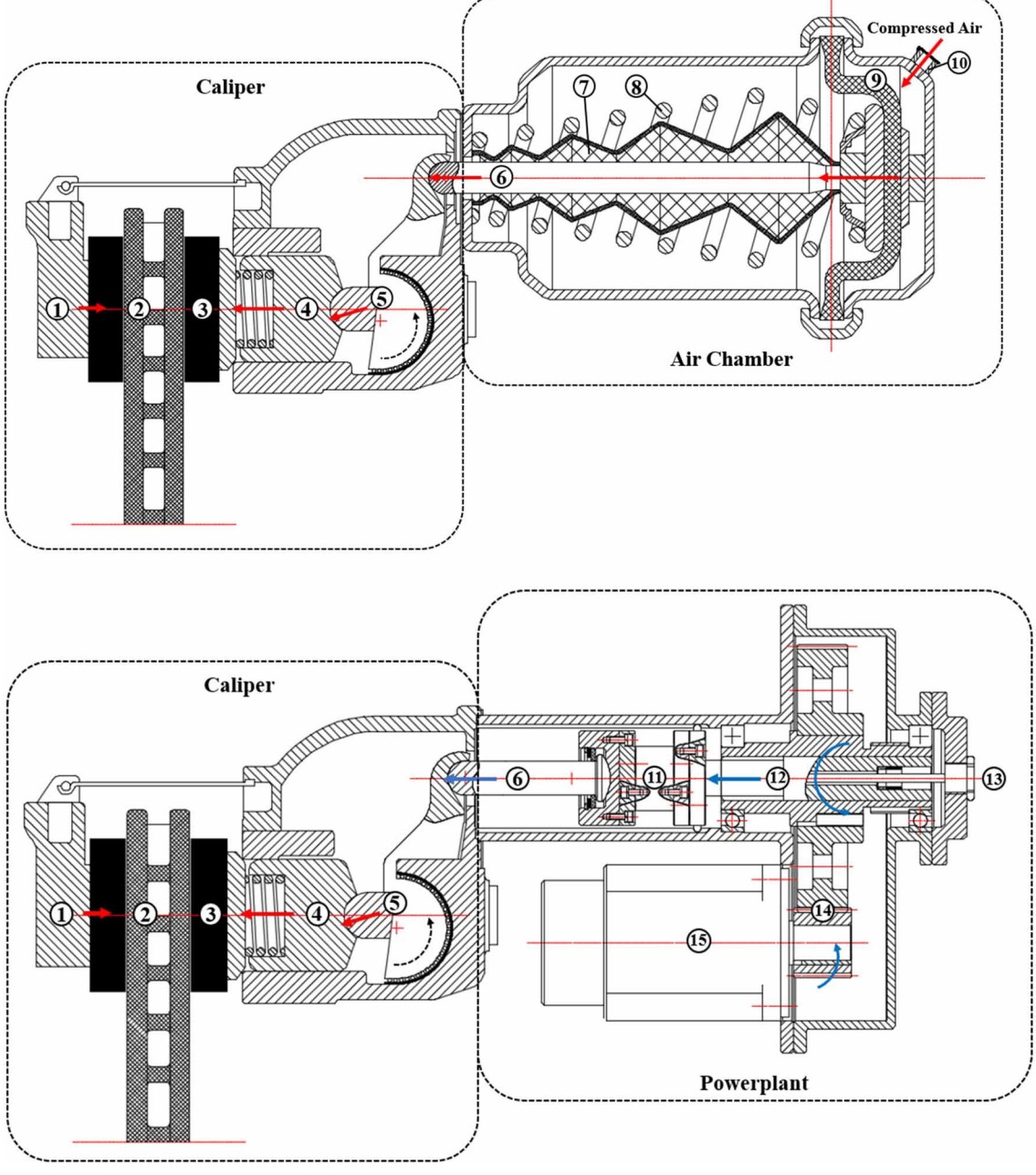

**Fig 1. Structures of the pneumatic–disc–brake and the EMB.** (a) Structure of the pneumatic–disc–brake; (b) Structure of the improved EMB.

mechanism to adjust the brake clearance. Therefore, the caliper is still retained in the improved EMB.

Particularly, to retain the caliper, the brake chamber is replaced by the powerplant to push the mechanisms inside the caliper that generates the braking pressure [3]. While, in the proposed EMB, the pressure pushing the push rod (6) is generated by a motor (15), a reduction mechanism (14) and a screw mechanism (12), and the power transmission path of the EMB in the powerplant is shown with blue arrows in Fig 1B. During the working of the system, the

torque output from the motor is amplified by the reduction mechanism (14) and transformed by the screw (12) into an axial thrust force on the push rod (6) and act on the arm (5) in the caliper. The transmission path of the axial thrust force in the caliper is exactly the same as the pneumatic–disc–brake shown in Fig 1A. Furthermore, the pressure sensor (11) measures the axial thrust and the linear sensor (13) measures the distance axially traveled by the screw.

Compared with the previous scheme, the improved EMB reduces the requirements of motor performances attribute to the reduction mechanism.

There are two functions of the caliper: the one is to amplify the axial thrust output from the powerplant and transform it into the braking pressure. Simultaneously, the caliper applies the reaction force between the brake shoes and the disc to the powerplant. At this point, the caliper acts as a "load" to balance the movement of the whole EMB.

## 2.2 EMB system model

The EMB system model is divided into two parts: the powerplant model and the caliper model. Where, the powerplant model is fabricated according to the electrical fundamentals and the kinematics. Please refer to [3] for specific process.

If the phase currents $i_a$, $i_b$, $i_c$ and the rotor mechanical angular speed $\omega_m$ are selected as the system state variables, the phase voltages $u_a$, $u_b$, $u_c$ and the load $F_L$ as the inputs, and the braking pressure $F_p$ and the screw axial velocity $v_x$ as the outputs, then the caliper model can be expressed as follows:

$$\begin{cases} \dot{x} = Ax + Bu \\ y = Cx \end{cases}, \tag{1}$$

where,

$$x = (i_a, i_b, i_c, \omega_m)^T, \ y = (F_p, v_x)^T, \ u = (u_a, u_b, u_c, F_L)^T,$$

$$A = \left( \begin{array}{ccc|c} -R/(L-M) & 0 & 0 & -p\psi_m f_a(\theta_m)/(L-M) \\ 0 & -R/(L-M) & 0 & -p\psi_m f_b(\theta_m)/(L-M) \\ 0 & 0 & -R/(L-M) & -p\psi_m f_c(\theta_m)/(L-M) \\ \hline \eta p\psi_m g \cdot f_a(\theta_m)/J & \eta p\psi_m g \cdot f_b(\theta_m)/J & \eta p\psi_m g \cdot f_c(\theta_m)/J & -D/J \end{array} \right),$$

$$B = \left( \begin{array}{ccc|c} 1/(L-M) & 0 & 0 & 0 \\ 0 & 1/(L-M) & 0 & 0 \\ 0 & 0 & 1/(L-M) & 0 \\ \hline 0 & 0 & 0 & -d_m/2kJ \end{array} \right),$$

$$C = \left( \begin{array}{ccc|c} 2\pi\eta pk\psi_m g \cdot f_a(\theta_m)/l & 2\pi\eta pk\psi_m g \cdot f_b(\theta_m)/l & 2\pi\eta pk\psi_m g \cdot f_c(\theta_m)/l & 0 \\ \hline 0 & 0 & 0 & l/2\pi \end{array} \right),$$

where, $R$ is the phase resistance, $L$ and $M$ are the self and mutual inductance of the phase, $p$ is the pole pairs of the motor, $\psi_m$ is the maximum winding flux, $f(\theta_m)$ is the back EMF waveform, $\eta$ is the efficiency of the screw, $D$ is the viscous damping coefficient of the motor, $J$ is the equivalent moment inertia to the rotor, $l$ and $d_m$ are the lead range and the nominal diameter of the

screw, and $k$ is the amplifying coefficient of the caliper, $g$ is the ratio of the reduction mechanism.

The caliper model is complex and described by the characteristics provided by the manufacturer to ensure the accuracy. The characteristics describe the relationship between the screw axial displacement $x_a$ and the normal pressure $F_N$ on the disc. Fig 2 shows the caliper model, where Fig 2A is the pneumatic–disc–brake based test data using a 19.5 inch chamber provided by the manufacturer. Fig 2B is the caliper model available for the proposed EMB after unit conversion (from MPa to kN) and fitting based on Fig 2A.

The EMB system model is shown in Fig 3. The upper part of the figure is the powerplant model established according to the state–space (1), and the lower part is the caliper model described in Fig 2. Where, the inputs of the powerplant model are the three–phase voltages $u_x$ and the load $F_L$ ($F_L = F_N/k$), and the outputs are the braking pressure $F_P$ and the screw axial velocity $v_x$.

### 2.3 Control prototype

**2.3.1 Overall control architecture: Distributed and layered.** In general, the overall control architecture adopts a distributed and layered structure composed of an EMB central unit (ECU) and several wheel braking pressure control units (WCUs), and the WCUs are distributed at each wheel. The ECU demonstrates a three–layered structure: the upper layer determines the total braking pressure according to the vehicle's dynamic states, the wheels rolling states, the brake pedal travels and other signals; the middle layer determines the target braking pressure in each wheel based on the I–curve; the lower layer is composed of an anti–lock braking system (ABS) module, which can avoid wheel locking by correcting the target braking pressure appropriately when the wheel has the locking tendency.

The WCUs also demonstrates a three–layered structure, and realizes the adjustment of the braking pressure for each wheel. The upper layer receives the target braking pressure command from the ECU and determines the target motor current by the braking pressure and motor speed adjustment modules. The middle layer compensates the friction of the screw, and the lower layer drives the motor to adjust the wheel braking pressure. The overall control architecture is shown in Fig 4.

**2.3.2 ECU: Determines and distributes the total braking pressure.** In this study, there is a linear relationship between the total braking pressure and the brake pedal travel, and the maximum pedal travel corresponds to the maximum braking pressure.

Since the braking pressure in the EMBs can be adjusted independently, the total braking pressure is distributed based on the I–curve. When the vehicle braking in the straight way, the braking pressure in each axle are determined by the following [19]:

$$F_{p.2} = \frac{1}{2}\left[\frac{G}{h_g}\sqrt{b^2 + \frac{4h_g L_a}{G}F_{p.1}} - \left(\frac{Gb}{h_g} + 2F_{p.1}\right)\right],\tag{2}$$

where, $F_{p.1}$ and $F_{p.2}$ are the braking pressures in the front and the rear axle, $G$ is the vehicle gravity, $h_g$ is the centroid height, $L_a$ is the wheelbase, and $b$ is the distance between the centroid and the rear axle center.

**2.3.3 ECU: ABS control.** *(1) ABS working principle*. The ABS is to identify the states of the wheels, to avoid the wheels locking by adjusting the wheel braking pressure in proper time, and to ensure braking stability of the vehicle. In the traditional braking systems, the ABS is realized by adjusting the pressure in the braking pipe through a solenoid valve. While, due to the braking pressure of each wheel is controlled independently, the ABS based on EMBs is more flexible.

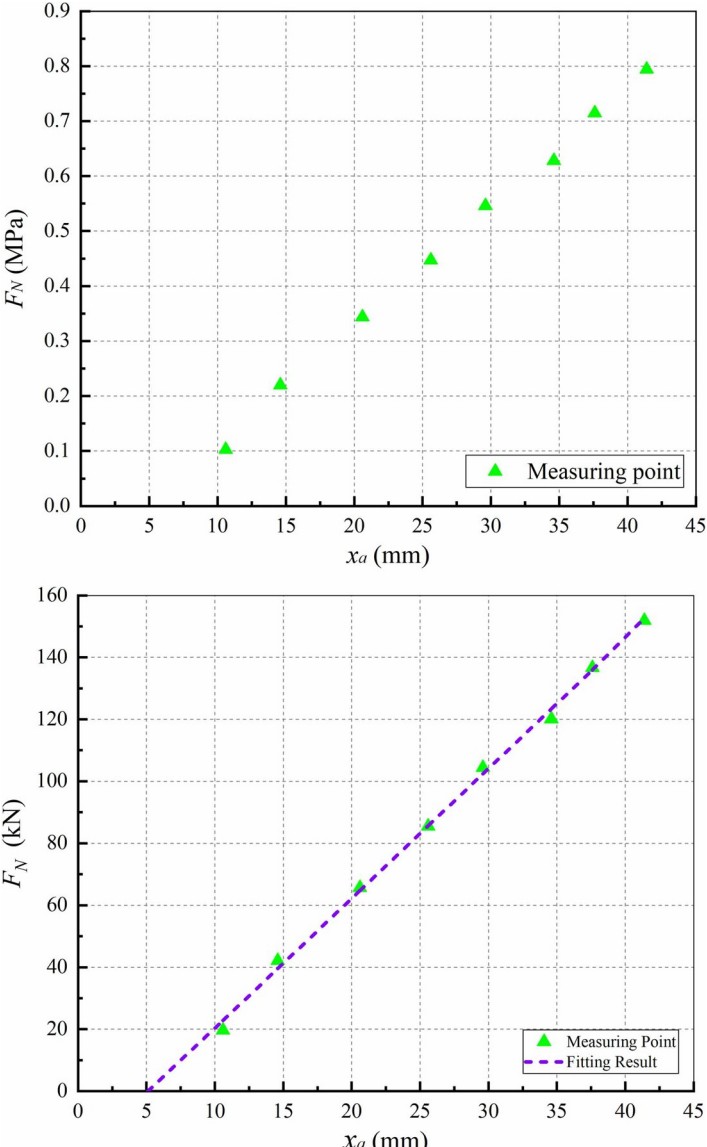

**Fig 2. Caliper model.** (a) The caliper characteristics provided by the manufacturer; (b) The fitted caliper model after unit conversion.

A crucial index to measure the wheel states is the slip ratio $\lambda$, which indicates the proportion of wheel slip in pure rolling when braking. The slip ratio $\lambda$ is defined as:

$$\lambda = \frac{v - \omega_w r_r}{v} \times 100\% \tag{3}$$

where, $\omega_w$ is the wheel angular velocity, $r_r$ is the wheel rolling radius, and $v$ is the velocity in the wheel center.

The slip ratio $\lambda$ determines the longitudinal force $F_{xb}$ and the lateral force $F_y$ acting on the wheels. Fig 5 shows the relationships between the longitudinal force $F_{xb}$ and the lateral force $F_y$ with the slip ratio $\lambda$. It can be seen from the figure that the lateral force $F_y$ decreases with the slip ratio $\lambda$. When the slip ratio $\lambda$ reaches 100%, the lateral force $F_y$ is 0, and the vehicle will

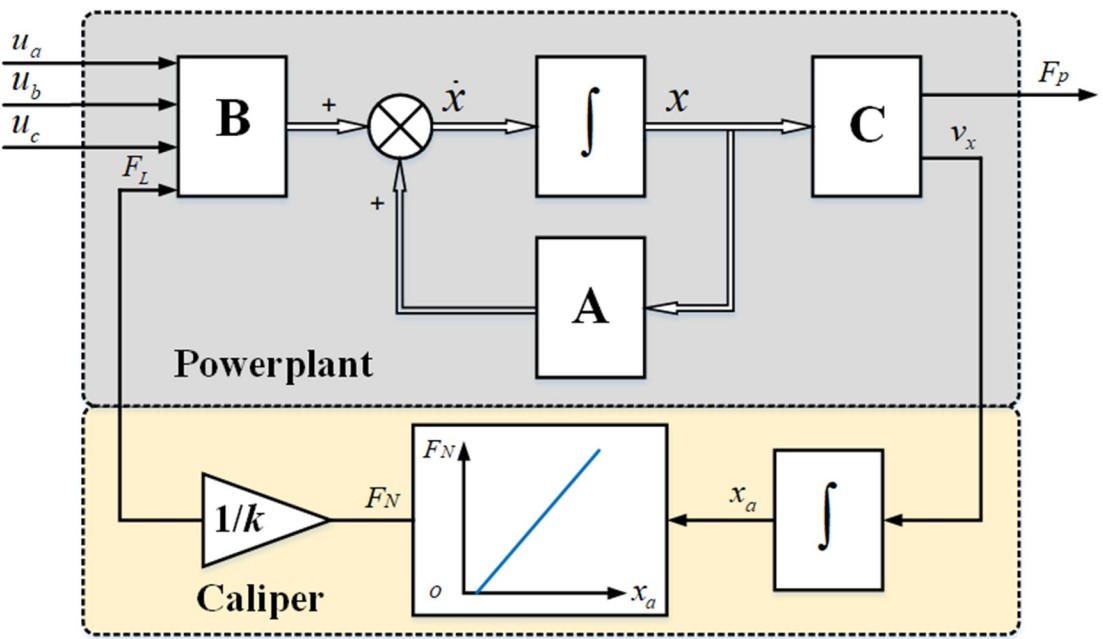

**Fig 3. EMB system model.**

lose the steering ability or fall into the risk of tail flick. At the same time, the longitudinal force $F_{xb}$ increases rapidly with the slip ratio $\lambda$ at the beginning, when the slip ratio $\lambda$ reaches to a certain range, the longitudinal force $F_{xb}$ gradually decreases with the slip ratio $\lambda$.

To ensure the wheels are subjected to a greater braking force and a certain lateral force which significant to the vehicle's lateral stability, the ABS should keep the slip ratio $\lambda$ of each wheel near the optimal slip ratio $\lambda^*$. The test data show that the optimal slip ratio $\lambda^*$ of pneumatic tire is about 15% ~ 20% when driving on a rigid road [19]. In this paper, the optimal slip ratio $\lambda^*$ is set to 20%.

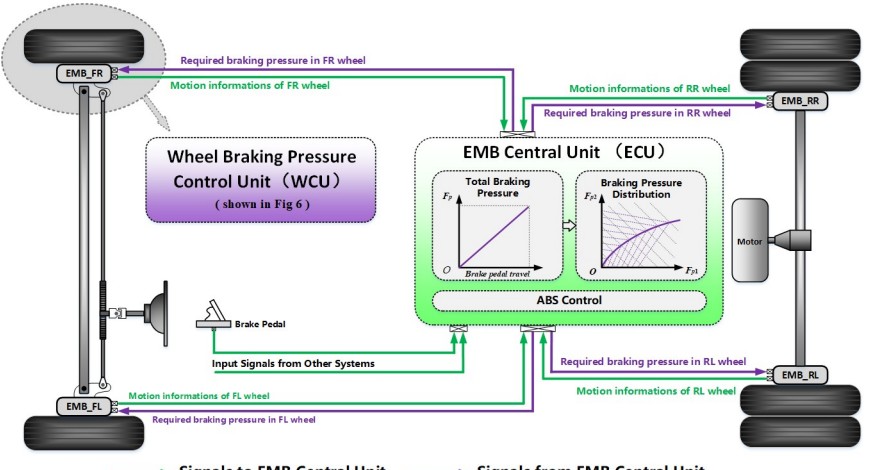

**Fig 4. Overall architecture of the control prototype.**

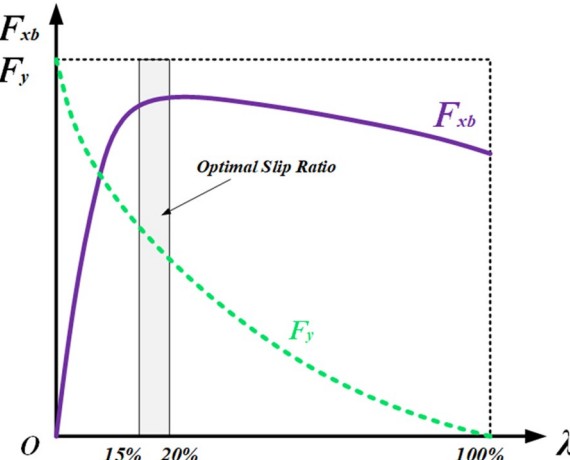

**Fig 5. The relationships between the $F_{xb}$ and the $F_y$ with the $\lambda$.**

*(2) ABS control law.* The sliding mode control (SMC) is an appropriate way to solve nonlinear systems. It demonstrates the characteristics of fast response, insensitive to parameter changes and external disturbances. This section deduces the control law of the wheel braking pressure when the ABS is triggered based on the single–wheel model and the SMC. The single–wheel model is shown with Eqs (4) and (5):

$$J_w \dot{\omega}_w = r_r F_{xb} - T_b, \tag{4}$$

$$F_{xb} = \mu F_z, \tag{5}$$

where, $J_w$ is the wheel moment inertia, $F_{xb}$ and $T_b$ are the longitudinal force and the braking torque on the wheel, $\mu$ is the adhesion coefficient between the tire and the ground, and $F_z$ is the vertical load on the wheel.

The ABS aims to keep the wheel slip ratio $\lambda$ near the optimal ratio $\lambda^*$ by adjusting the braking pressure. In order to eliminate the tracking error of the slip ratio, the switching surface can be defined as follows:

$$s = \lambda - \lambda^*, \tag{6}$$

To short the adjustment time and attenuate the chatters, the reaching law is constructed combining the constant rate and power reaching laws [20]:

$$\dot{s} = -k_1 \, \text{sign}(s) - k_2 |s|^\alpha \, \text{sign}(s), \tag{7}$$

where, $k_1$, $k_2$ and $\alpha$ are the SMC parameters and there are $k_1$, $k_2 > 0$ and $0 < \alpha < 1$. Obviously, there is $s\dot{s} < 0$, and the reaching law satisfies the stability requirements of the SMC.

Assuming that the road conditions remain unchanged, the braking pressure control law of the optimal slip ratio can be derived from the Eqs (3)–(7):

$$F_p = \frac{\mu r_r F_z}{2\mu_b r_b} - \frac{J_w}{2\mu_b r_b r_r}\left[(k_1 + k_2|s|^\alpha)\text{sign}(s)v + (1-\lambda)\frac{dv}{dt}\right], \tag{8}$$

where, $\mu_b$ is the friction coefficient between the brake shoes and the disc, and $r_b$ is the effective radius of the disc.

In the mentioned control law, the larger of the $k_1$ and $k_2$ values, the faster of the $F_p$ convergence, and the shorter time to the optimal slip ratio $\lambda^*$. While, the chatters of the $F_p$ maybe caused if the values are too large. Furthermore, in the power reaching law, the $\alpha$ is to adjust the convergence rate of the $F_p$ based on the error between the $\lambda$ and the $\lambda^*$. Hence, the larger of the $\alpha$, the slower of the $F_p$ convergence under the smaller error, which can reduce the chatters of the $F_p$ appropriately.

It should be noted that the ABS control prototype is not activated when the vehicle velocity is lower than 5 km/h.

**2.3.4 WCU: Wheel braking pressure control.** The braking pressure control prototype is integrated in the WCUs, which also a three–layer structure. The upper layer is composed of a braking pressure and a motor speed adjustment module, and the middle layer is a friction compensation module, and the lower layer is a motor drive module, as showed in Fig 6.

*(1) Upper layer*: Realizes the adjustments of the braking pressure and the motor speed. The tasks of this layer are to determine the target motor speed and current according to the braking pressure commands from the ECU, and the tasks are implemented by the PID algorithm:

$$u(t) = k_p e(t) + k_i \int_0^t e(t)dt + k_d \frac{de(t)}{dt}, \tag{9}$$

where, $u(t)$ is the PID output, $e(t)$ is the PID input and $k_p$, $k_i$ and $k_d$ are the PID parameters. Specifically, the input of the braking pressure adjustment module is the error between the target braking pressure $F_{p\_target}$ and the actual pressure $F_p$, and the output of this module is the target motor speed $n_{target}$. Similarly, the motor speed adjustment module takes the error between the target motor speed $n_{target}$ and the actual speed $n$ as input, and determines the target motor current $i_{target}$ by adjusting the error.

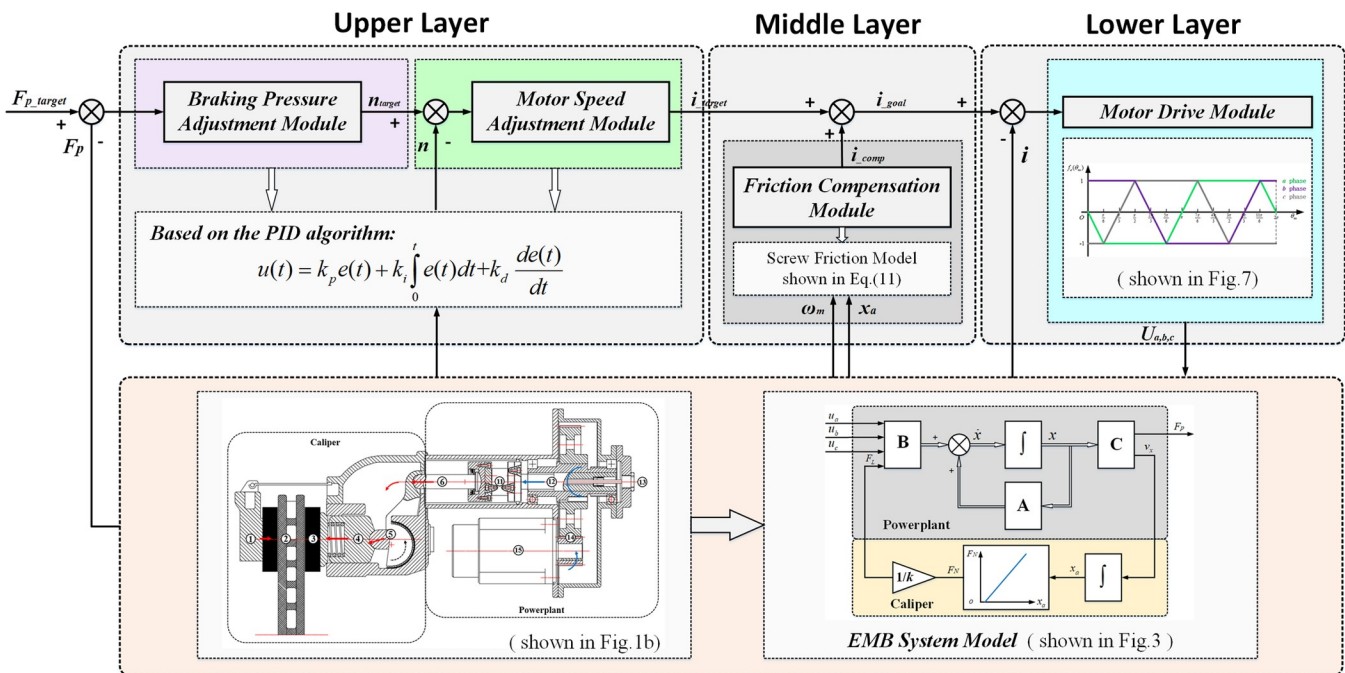

**Fig 6. Wheel braking pressure control prototype.**

In the PID algorithm, the $k_p$ is set to accelerate the adjustment and reduce the steady–state error. While, the overshoots of the $u(t)$ maybe increased and the adjustment process maybe prolonged if the $k_p$ is too large. Moreover, the $k_i$ is set to eliminate the steady–state error. While, the response speed of the $u(t)$ maybe affected and the overshoots of the $u(t)$ maybe increased. Finally, the role of the $k_d$ is attenuating the overshoots appropriately and improving the dynamic performance of the $u(t)$.

*(2) Middle layer*: *Realizes the friction compensation*. To compensate the nonlinear friction of the screw, the friction compensation module is introduced into the WCUs. The module is used to make the motor produce an additional torque to overcome the screw friction by determining a compensation current $i_{comp}$. When the motor operates in the $2\pi/3$ conduction mode, and the transient process of commutation is ignored, then the compensation current is:

$$i_{comp} = \frac{T_f}{K_T} = \frac{F_f \cdot d_m}{4p\psi_m},$$ (10)

where, $T_f$ is the motor torque to overcome the friction, $F_f$ is the screw friction, and can be calculated by the LuGre's friction model described below [21]:

$$\begin{cases} F_f = \sigma_0 z + \sigma_1 \dot{z} + \sigma_2 v_T \\ \dot{z} = v_T - \sigma_0 \dfrac{|v_T|}{g(v_T)} z \\ g(v_T) = F_c + (F_s - F_c)e^{-(v_T/v_s)^2} \end{cases},$$ (11)

where, $\sigma_0$ is the rigidity coefficient, $\sigma_1$ and $\sigma_2$ are the viscous damping and friction coefficients respectively, $g(v_T)$ is the Stribeck's function, $F_c$ and $F_s$ are the Coulomb and static frictions respectively, $v_s$ is the Stribeck's velocity. Because of the surface contact between the screw and the head, the relative velocity $v_T$ is:

$$v_T = \omega_m \frac{d_m}{2} \cdot \cos(\zeta + \rho) - \dot{x}_a \sin(\zeta + \rho),$$ (12)

where, $x_a$ is the screw axial travel, $\zeta$ and $\rho$ are the helix and equivalent friction angles of the screw.

*(3) Lower layer*: *Drives the motor*. The motor drive module is composed of a hysteresis controller and an inverter. The module receives the goal current demand $i_{goal}$ ($i_{goal}$ is the sum of $i_{target}$ and $i_{comp}$), and generates pulses through the hysteresis controller to turn on the corresponding power switching of the inverter. Where, the inverter operates with the three–phase bridge and two–way conduction mode. For the principle of hysteresis controller, please refer to [22–25].

## 2.4 Model parameters

The parameters of the EMB system model are shown in Table 1.

The back EMF waveform of motor is shown in Fig 7.

The relevant parameters of the vehicle are shown in Table 2.

Where, $m$ is the full load mass of the vehicle, and $a$ is the distance between the centroid and the front axle center, and the rest symbols have the same meaning as before. The preliminary parameters of the control prototype are shown in Table 3.

**Table 1. Parameters of the EMB system model.**

| Category | Abbreviation | Value | Unit | Category | Abbreviation | Value | Unit |
|---|---|---|---|---|---|---|---|
| Motor related | $R$ | 0.5 | Ω | Screw related | $d_m$ | 28 | mm |
| | $L$ | $4.74 \times 10^{-3}$ | H | | $l$ | 5 | mm |
| | $M$ | $2.93 \times 10^{-3}$ | H | | $\eta$ | 0.5 | - |
| | $p$ | 3 | - | | $\zeta$ | 3.57 | degree |
| | $\psi_m$ | $7 \times 10^{-3}$ | Wb | | $\rho$ | 4.5739 | degree |
| | $D$ | $1.5 \times 10^{-3}$ | N·m/rpm | Others | $g$ | 4 | - |
| | $J$ | $1.13 \times 10^{-3}$ | kg·m$^2$ | | $k$ | 15.2 | - |

## 3. Multi–objectives optimization of the EMB parameters

This section aims to improve the vehicle's braking performance under the constraints of relevant regulations by optimizing the EMB parameters based on the NSGA–II algorithm. First, the objectives and constraints are set. Subsequently, the decision variables are selected and the optimal mathematical model is established. Finally, the co–simulation platform based on Matlab/Simulink and TruckSim is built.

### 3.1 Optimal objectives

In general, the vehicle's braking performance is evaluated by the braking efficiency, the efficiency constancy and the braking stability. Specifically, the braking efficiency mainly includes the stopping distance and the deceleration. The efficiency constancy refers to the ability of a brake to resist thermal recession. The braking stability means the vehicle without deviation, sideslip and loss of steering when braking.

With reference to ECE R13 [26], GB 12676–2014 [27] and GB 7258–2017 [28], the braking efficiency and stability requirements of the M3 vehicles are shown in Table 4:

Combining with Table 4 and [29], the stopping distance (SD), the mean fully developed deceleration (MFDD) and the maximum lateral displacement of the body (Max LD) are set as optimal objectives in this study.

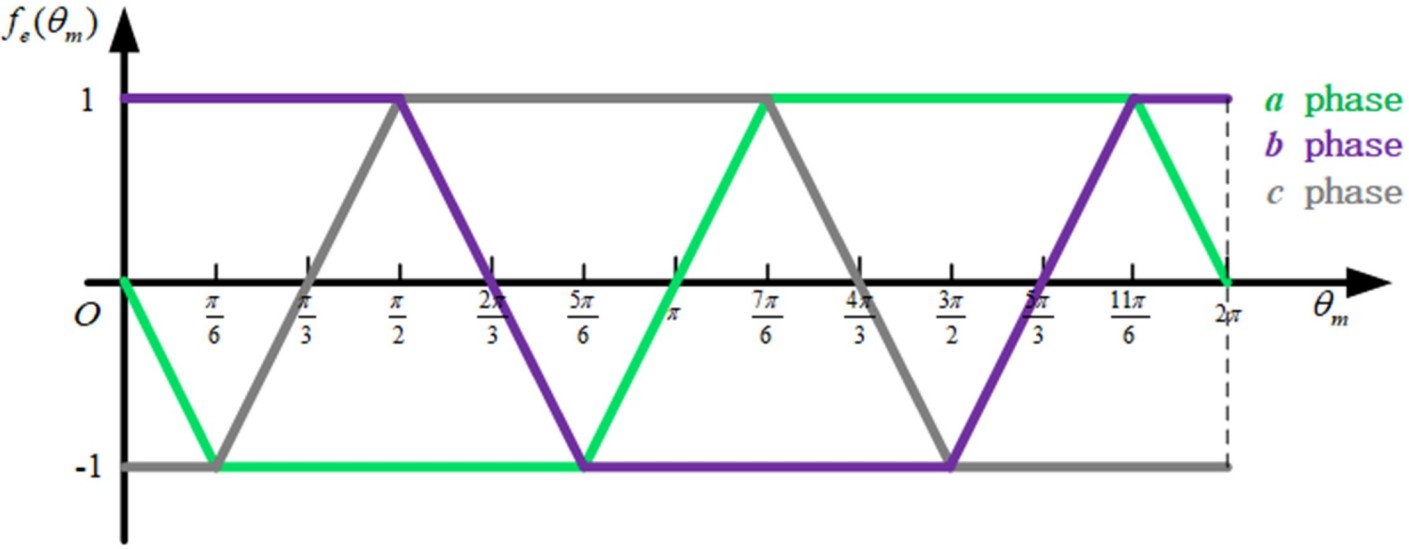

**Fig 7. Back EMF waveform of motor.**

**Table 2. The relevant parameters of the vehicle.**

| Category | Abbreviation | Value | Unit | Category | Abbreviation | Value | Unit |
|---|---|---|---|---|---|---|---|
| Body related | $m$ | 14300 | kg | Wheel related | $r_r$ | 0.4815 | m |
| | $h_g$ | 0.53 | m | | $r_b$ | 0.173 | m |
| | $L_a$ | 6.12 | m | | $\mu_b$ | 0.4 | - |
| | $a$ | 3.2 | m | | $J_w$ | 14 | kg·m$^2$ |

**3.1.1 Stopping Distance (SD).** The SD is one of the important indexes of the braking efficiency, and it can be reflected the continuous output capacity of a braking system. The theoretical formula of the SD is shown below:

$$s = (t + \frac{t'}{2})u_0 + \frac{u_0^2}{2a_{b\max}}, \tag{13}$$

where, $t$ is the time to eliminate the brake clearance, $t'$ is the time that the braking pressure to the target value, $u_0$ is the initial velocity, and $a_{b\max}$ is the maximum deceleration of the vehicle.

**3.1.2 Mean Fully Developed Deceleration (MFDD).** The MFDD is another index of the braking efficiency. To evaluate the braking efficiency using the MFDD instead the instantaneous deceleration can avoid the inaccurate results caused by the fluctuations of the latter. Moreover, it also eliminates the interferences caused by the driver's reaction and the brake clearance elimination stages. Where, the driver's reaction stage is defined as the process from receiving the emergency braking demand to hitting the brake pedal. Actually, no matter in the driver's reaction or the brake clearance elimination stage, the wheels are not affected by the braking pressure.

The MFDD is calculated as follows:

$$\text{MFDD} = \frac{(u_b^2 - u_e^2)}{25.92(s_e - s_b)}, \tag{14}$$

where, $u_0$ is the initial velocity, $u_b$ equals to $0.8u_0$, $u_e$ equals to $0.1u_0$, $s_b$ is the distance from $u_0$ to $u_b$, and $s_e$ is the distance from $u_0$ to $u_e$.

**3.1.3 Maximum lateral displacement of the body (Max LD).** To evaluate the braking stability more intuitive, the lateral displacement of the body (LD) is introduced into the optimal objectives. In this study, the following coordinate is used to define the sign of the LD: the positive direction of the $x$ axis points to the vehicle's forward, and the positive direction of the $y$ axis points to the left side of the forward. Hence, if the body deviates to the left, the LD is positive; otherwise, it is negative under this coordinate.

It should be noted that the above definition can be used to determine the vehicle's deviational direction during the braking. While, the LD is taken as the absolute value to facilitate the calculations in the optimization.

**Table 3. The preliminary parameters of the control prototype.**

| Category | Abbreviation | Value | Category | Abbreviation | Value |
|---|---|---|---|---|---|
| Braking pressure adjustment module in the WCUs | $k_{p\_F}$ | 1500 | ABS control prototype in the ECU | $k_1$ | 6 |
| | $k_{i\_F}$ | 0.15 | | $k_2$ | 8 |
| | $k_{d\_F}$ | 0.1 | | $\alpha$ | 0.1 |
| Motor speed adjustment module in the WCUs | $k_{p\_n}$ | 1.5 | | | |
| | $k_{i\_n}$ | 0.15 | | | |
| | $k_{d\_n}$ | 0.01 | | | |

**Table 4. The braking efficiency and stability requirements of the M3 vehicles.**

| Performance requirements | Indicators and requirements | | Relevant regulations |
|---|---|---|---|
| | Stopping distance | $\leq$75.57m | GB 12676–2014 |
| | Mean fully developed deceleration | $\geq$4.0 m /s$^2$ | GB 12676–2014 |
| | Braking stability | No part of the vehicle deviates from the 3 m wide road. | GB 7258–2017 |
| Test conditions | Test road condition | Flat dry concrete or asphalt pavement with adhesion coefficient greater than 0.7. | |
| | Load state | Full loads | |
| | Initial velocity | 90 km/h | |

## 3.2 Decision variables

There are many parameters of the EMB system model according to Fig 3 and Table 3. These parameters demonstrate different effects on the system performance. If all the parameters are considered as decision variables, the computational burden will be increased. Hence, this section first analyzes the sensitivity of parameters to the system based on the orthogonal design, and then selects the parameters which have great effect on the EMB performance as decision variables [30].

**3.2.1 Parameters coupling relations.** According to the EMB system model and the control prototype, the parameters coupling relations are sorted out and shown in Fig 8.

**3.2.2 Parameters sensitivity.** To determine the effect of each parameter on the EMB performance, the parameters sensitivity analyses are performed, which are based on the orthogonal design. Before the design, the levels of each parameter in Table 1 was set, and then the appropriate orthogonal table was selected according to the parameters and levels. During the design, the maximum braking pressure and the response time of the single EMB were taken as

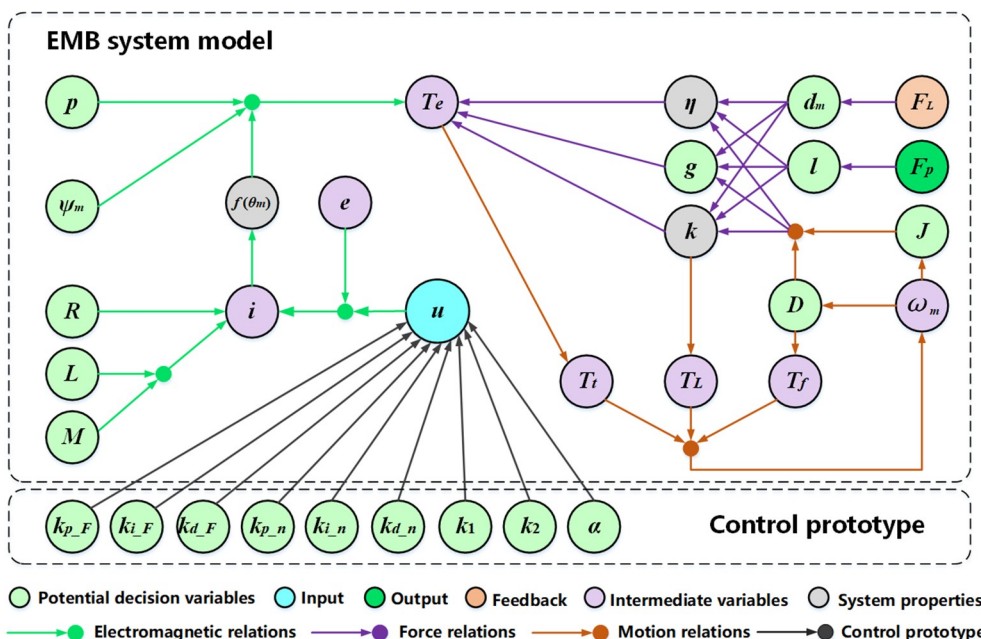

**Fig 8. Parameters coupling relations.** The coupling relations in Fig 8 include the electromagnetic relations, the force relations, the motion relations and the control prototype. Where, the green labeled parameters are the potential decision variables, the gray parameters are the EMB attributes, the purple parameters are the intermediate variables during calculation, and the rest are the input, the output and the feedback of the model.

the indexes, and the results under different levels were recorded. The orthogonal design was only conducted for the structural parameters.

*(1) Parameters and levels*. The structural parameters and their levels determined according to Table 1 are shown in Table 5:

*(2) Orthogonal design and range analyses*. The $L_{27}(3^{13})$ orthogonal table was selected and the orthogonal design with 10 factors and 3 levels was performed according to the above parameters and levels. The results are shown in S1 Appendix and the range analyses are shown in S2 Appendix.

In S2 Appendix, the $K_n$ ($n$ = 1, 2, 3) refers to the sum of recorded indexes corresponds to the $n^{th}$ level of each parameter, and the $\bar{K}_n$ is average of the $K_n$. The range is defined as follows:

$$R_a = \max\{\bar{K}_n\} - \min\{\bar{K}_n\},\qquad(15)$$

Hence, the range of each parameter relative to the recorded indexes of the maximum braking pressure and response time are shown in Table 6.

*(3) Parameters sensitivity*. The parameters sensitivity ranking can be obtained according to the range analyses, as shown in Table 7.

**3.2.3 Decision variables.** From the above analyses, it can be seen that the parameters $g$, $l$ and $j$ have great effect on the response time, and their ranges are more than 1; while the parameters $\psi_m$, $D$ and $P$ mainly affect the maximum brake pressure, and their ranges are more than 10. Therefore, the structural parameters such as $g$, $l$, $J$, $\psi_m$, $D$ and $P$, and the control parameter such as $k_{p\_F}$, $k_{i\_F}$, $k_{d\_F}$, $k_1$, $k_2$ and $\alpha$ are selected as the decision variables.

## 3.3 Constraints

**3.3.1 Equality constraints: The value bounds of decision variables and the model parameters.** Randomly assign the decision variable values may prolong the convergence process of results, and the results may not converge even the calculation may be terminated due to the parameters mismatches. To avoid the phenomenon, the decision variable values should be limited to a certain range according to the model parameters listed in Section 2.4, and as shown in Table 8.

Furthermore, the EMB attributes ($k$ = 15.2, $\eta$ = 0.5) and the parameters other than decision variables (such as $R$, $L$, $M$, $d_m$, $k_{p\_n}$, $k_{i\_n}$ and $k_{d\_n}$) described in Fig 8 will also set as constraints.

**3.3.2 Inequality constraints: Constraints of the EMB performance.** The constraints of the EMB performance mainly include two aspects: one is the minimum requirements of a braking systems defined by the relevant regulations, as shown in Table 4. Moreover, the maximum braking torque should not be lower than the maximum braking torque provided by the ground to ensure the vehicle's sufficient braking reserve.

The vehicle's load will transfer to the front axle during braking. Hence, the braking torque requirements of the front axle will be greater than the rear. Therefore, the constraint of

**Table 5. Parameters and levels.**

| Parameter | Levels | | | Unit | Parameter | Levels | | | Unit |
|---|---|---|---|---|---|---|---|---|---|
| | 1 | 2 | 3 | | | 1 | 2 | 3 | |
| $R$ | 0.4 | 0.5 | 0.6 | Ω | $g$ | 3 | 4 | 5 | - |
| $L$ | $4.5 \times 10^{-3}$ | $4.8 \times 10^{-3}$ | $5 \times 10^{-3}$ | H | $d_m$ | 25 | 30 | 35 | mm |
| $M$ | $2 \times 10^{-4}$ | $2.5 \times 10^{-4}$ | $3 \times 10^{-4}$ | H | $l$ | 4 | 5 | 6 | mm |
| $p$ | 3 | 5 | 8 | - | $J$ | $1 \times 10^{-3}$ | $2 \times 10^{-3}$ | $3 \times 10^{-3}$ | kg m$^2$ |
| $\psi_m$ | 0.002 | 0.007 | 0.012 | Wb | $D$ | 0.001 | 0.002 | 0.003 | N m/rpm |

**Table 6. Results of the range analyses.**

| Parameters | | R | L | M | p | $\psi_m$ | g | $d_m$ | l | J | D |
|---|---|---|---|---|---|---|---|---|---|---|---|
| Range | Response time | 0.082 | 0.026 | 0.027 | 0.042 | 0.047 | 0.307 | 0.062 | 0.202 | 0.134 | 0.068 |
| | Maximum braking pressure | 6.68 | 1.48 | 6.66 | 12.05 | 14.48 | 6.21 | 6.17 | 1.24 | 7.16 | 12.17 |

maximum braking torque considering the brake distribution is:

$$T_{b\max} \geq \frac{Mg\varphi_{\max}r_r(\varphi_{\max}h_g + b)}{2L_a}, \tag{16}$$

where, $T_{b\max}$ is the maximum braking torque, $\varphi_{\max}$ is the maximum ground adhesion coefficient.

### 3.4 Optimal mathematical model

The optimal mathematical model is as follows based on above analyses:

$$\min F(X) = [\text{SD}, \ -\text{MFDD}, \ |\text{Max LD}|],$$

$s.t.$

$$\begin{cases}
X = [g, l, J, \psi_m, D, p, k_{p\_F}, k_{i\_F}, k_{d\_F}, k_1, k_2, \alpha] \\
x_i^L \leq x_i \leq x_i^U, \ i = 1, 2, \ldots, 12 \\
\eta = 0.5 \\
k = 15.2 \\
R = 0.5 \\
L = 4.74 \times 10^{-3} \\
M = 2.93 \times 10^{-3} \\
d_m = 28 \\
k_{p\_n} = 1.5 \\
k_{i\_n} = 0.15 \\
k_{d\_n} = 0.01 \\
T_b \geq T_{b\max} \\
SD \leq SD_{\max} \\
MFDD \geq MFDD_{\min}
\end{cases} \tag{17}$$

**Table 7. Parameters sensitivity sorting.**

| | |
|---|---|
| **Sorting by the response time** | $g>l>J>R>D>d_m>\psi_m>p>M>L$ |
| **Sorting by the maximum braking pressure** | $\psi_m>D>p>J>R>M>g>d_m>L>l$ |

**Table 8. Value bounds of the decision variables.**

| Variables | | $g$ | $l$ | $J$ | $\psi_m$ | $D$ | $p$ | $k_{p\_F}$ | $k_{i\_F}$ | $k_{d\_F}$ | $k_1$ | $k_2$ | $\alpha$ |
|---|---|---|---|---|---|---|---|---|---|---|---|---|---|
| $i$ | | 1 | 2 | 3 | 4 | 5 | 6 | 7 | 8 | 9 | 10 | 11 | 12 |
| Minimum | $x_i^L$ | 3 | 4 | 0.001 | 0.005 | 0.001 | 2 | 1000 | 0.1 | 0.05 | 4 | 6 | 0.05 |
| Initial | $x_i^I$ | 4 | 5 | 0.00113 | 0.007 | 0.0015 | 3 | 1500 | 0.15 | 0.1 | 6 | 8 | 0.1 |
| Maximum | $x_i^U$ | 6 | 6 | 0.003 | 0.010 | 0.003 | 4 | 2000 | 0.2 | 0.15 | 8 | 10 | 0.15 |

### 3.5 Optimization algorithm

In this study, the NSGA–II is used to solve the optimization. The essence of the NSGA–II is genetic algorithm (GA), which generates new populations by selecting, crossing or mutating the individual chromosomes in the current population. Where, the chromosome is described as the carrier containing all the information of individuals. Moreover, an individual is actually a Pareto solution under the constraints, and the population is the set of all individuals, also a Pareto set.

The matrix shown in Fig 9 can be used to understand the relationship among chromosomes, individuals and population [31].

Different from the GA, the non–dominated sorting method and elitism mechanism are introduced in the NSGA–II. Specifically, the dominating relationships among individuals are judged and sorted by the method, then the Pareto fronts are formed and the Pareto ranks are gave to the individuals. The elitism mechanism is used to produce a new population by selecting the best individuals from the current population based on their Pareto ranks and crowding distances (CDs). Where, the CD is defined as follows [32]:

$$CD = \sum_{i=1}^{N} \sum_{k=2}^{K-1} |f_i(k+1) - f_i(k-1)|, \tag{18}$$

where, $f_i(x)$ is the $i^{th}$ objective value corresponds to the $x^{th}$ individual in the same Pareto front, $N$ is the dimension of the objectives, $K$ is the number of individuals in the current Pareto front. Particularly, the CD of the first and the last individuals in any Pareto fronts are defined as $\infty$.

The population evolution is realized based on the binary tournament selection, the simulated binary crossover and the polynomial mutation in this study. Where,

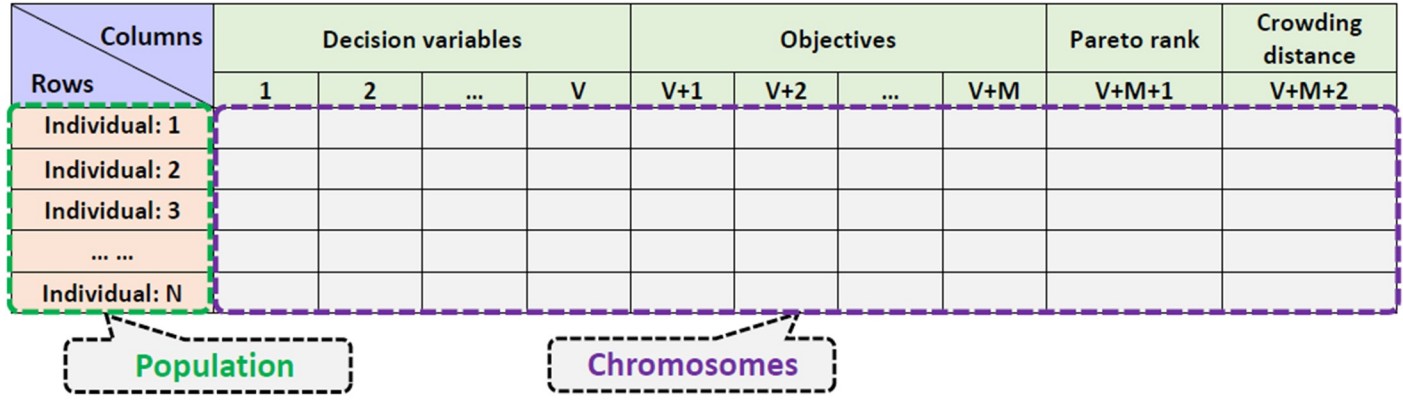

**Fig 9. Relationship among chromosomes, individuals and population.**

- The mechanism of the binary tournament selection is that two individuals are randomly selected from the population every time, and the better one is selected to participate in the evolution according to the principle of "low Pareto ranks and large CDs", and the remaining one is put back to the current population;

- For the principle of the simulated binary crossover, please refer to [31]. The method forms new individuals by exchanging part chromosomes of current individuals, and the implementation of the crossover is as follows:

$$
\begin{cases}
c_{1,k} = \dfrac{1}{2}(p_{1,k} + p_{2,k}) - \dfrac{1}{2}\left|\dfrac{c_{1,k} - c_{2,k}}{p_{1,k} - p_{2,k}}\right|(p_{2,k} - p_{1,k}) \\
c_{2,k} = \dfrac{1}{2}(p_{1,k} + p_{2,k}) + \dfrac{1}{2}\left|\dfrac{c_{1,k} - c_{2,k}}{p_{1,k} - p_{2,k}}\right|(p_{2,k} - p_{1,k})
\end{cases},
\tag{19}
$$

where, $p_{1,k}$ and $p_{2,k}$ are the $k^{\text{th}}$ decision variable values of the current individuals, and $c_{1,k}$ and $c_{2,k}$ are the values of the new individuals after the crossover.

- Similar to the crossover, the polynomial mutation also a method to generate new individuals. For its principle, please refer to [31], and the implementation of the mutation is as follows:

$$
c_k = p_k + (p_k^u - p_k^l)\delta_k,
\tag{20}
$$

where, $p_k^u$ and $p_k^l$ are the upper and lower bounds of the $k^{\text{th}}$ decision variable in the current individuals, $c_k$ is the decision variable after the mutation, and $\delta_k$ can be calculated by Eq (21):

$$
\begin{cases}
\delta_k = (2r_k)^{\frac{1}{\eta_m + 1}} - 1, \ r_k < 0.5 \\
\delta_k = 1 - [2(1 - r_k)]^{\frac{1}{\eta_m + 1}}, \ r_k \geq 0.5
\end{cases},
\tag{21}
$$

where, $r_k$ is a random number between 0 and 1, $\eta_m$ is the mutational distribution index.

The flow chart of NSGA–II is shown in Fig 10.

In this study, the population size is set to 50, the dimension of the objectives is 3, the dimension of the decision variables is 12, and the maximum evolutionary generation is set to 100. The initial population is randomly generated within the bounds of decision variables shown in Table 8. The probability of the gene crossover and the mutation are set to 0.9 and 0.1, respectively.

## 3.6 Co–simulation platform

A co–simulation platform for the EMB parameters optimization is built based on Matlab/Simulink and TruckSim, and shown in Fig 11.

In the platform, the NSGA–II is operated through the ".m" script in Matlab, while the braking pressure control prototype and the EMB system model are operated through the ".mdl" model in Simulink. There are two kinds of necessary parameters in the model operation: the first is the decision variables, which is generated after the NSGA–II running and transferred to Simulink by Matlab, and the value of these parameters are changed with the iterations. The rest are stored in the Simulink model workspace and remained constant during whole iteration.

The TruckSim model includes vehicle dynamics, road, driver and test procedures. After receiving the wheel braking pressure demands from Simulink, the vehicle motion statuses are calculated

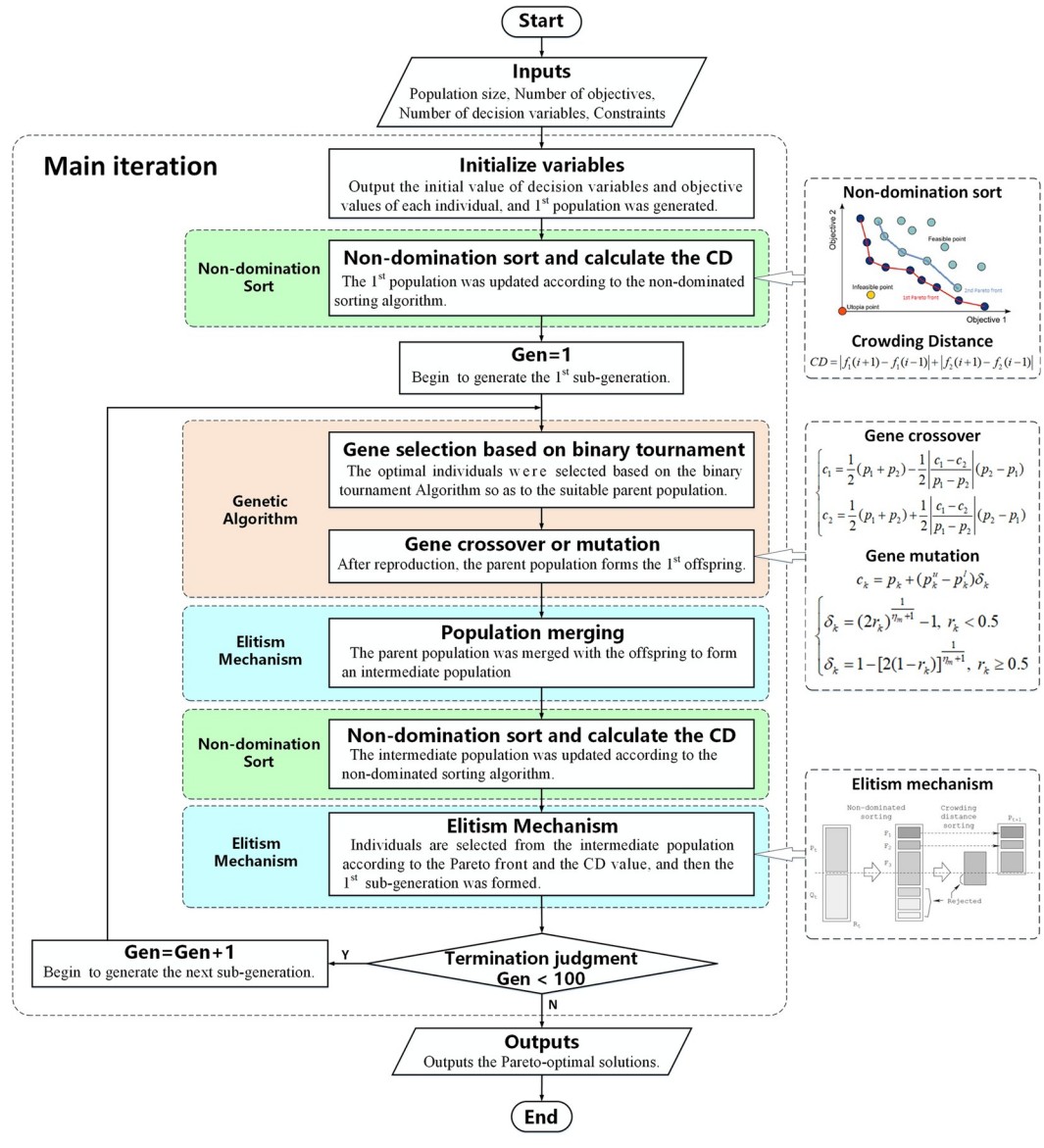

**Fig 10. The flow chart of NSGA–II.**

and fed back to the NSGA–II, the control prototype and the EMB system model to form a complete simulation process. The parameters required for the platform are described in Section 2.4.

## 4. Discussions and simulations

### 4.1 Optimal results

The simulation conditions are set to full load and straight road. Furthermore, the initial velocity is set to 90 km/h and the adhesion coefficient is 0.85. During the braking process, the driver holds the steering wheel tightly and does not correct the vehicle direction.

After 100 iterations, 50 individuals of a new population (also a Pareto set) are formed and located in the 1st Pareto front. The SD, the MFDD and the Max LD are set as the axes, and the MFDD ranges are represented by different colors, then the 50 individuals are expressed in Fig

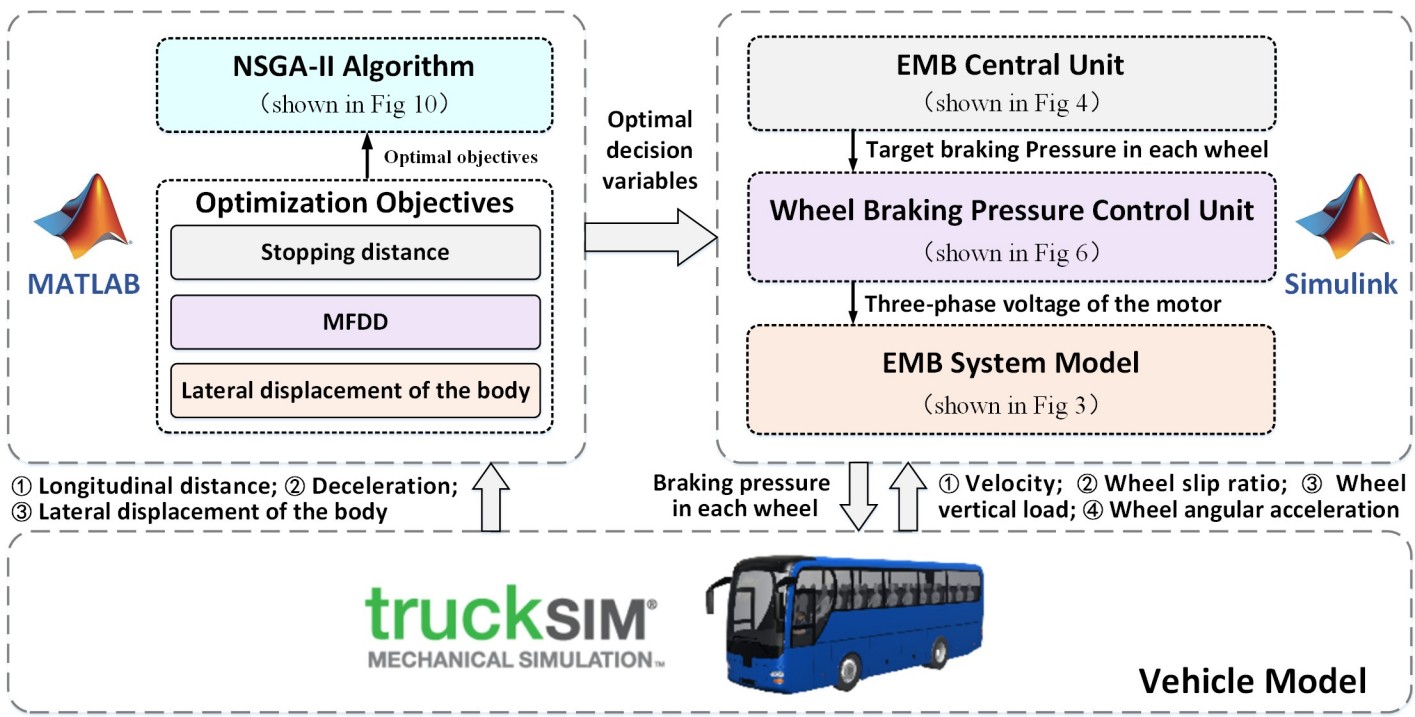

**Fig 11. Co–simulation platform for the EMB parameters optimization.**

12. To make the results intuitive, the numbers of the individuals are labeled, and the objective values correspond to each individual are projected to the coordinate planes respectively.

### 4.2 Discussions

It can be concluded from Fig 12 that:

- The SD of 90 km/h–0 on the straight and dry road is between 40.21 m and 46.92 m after optimization;

- The MFDD of the vehicle is between 0.878 g and 0.903 g during the whole braking;

- The Max LD is between 0.58 m and 0.592 m;

    The above indicators are satisfied with the requirements of the GB 12676–2014 and the GB 7258–2017. Where, ranges of the MFDD and the Max LD are smaller, and range of the SD is the largest. Hence, the MFDD range was divided into six intervals, and the SD, the MFDD, the Max LD, the Pareto rank and the CD of each individual were represented with radar maps, and as shown in Fig 13.

    The shortest SD was set as the main criterion, and the maximum MFDD and minimum Max LD were set as the auxiliary criteria to select the representative individual from each MFDD range referring to Fig 13:

- In the interval of MFDD ≥ 0.900 g, the 6[th] individual was selected;

- In the interval of 0.895 g ≤ MFDD < 0.900 g, the 4[th] individual was selected;

- In the interval of 0.890 g ≤ MFDD < 0.895 g, the 30[th] individual was selected;

- In the interval of 0.885 g ≤ MFDD < 0.890 g, the 40[th] individual was selected;

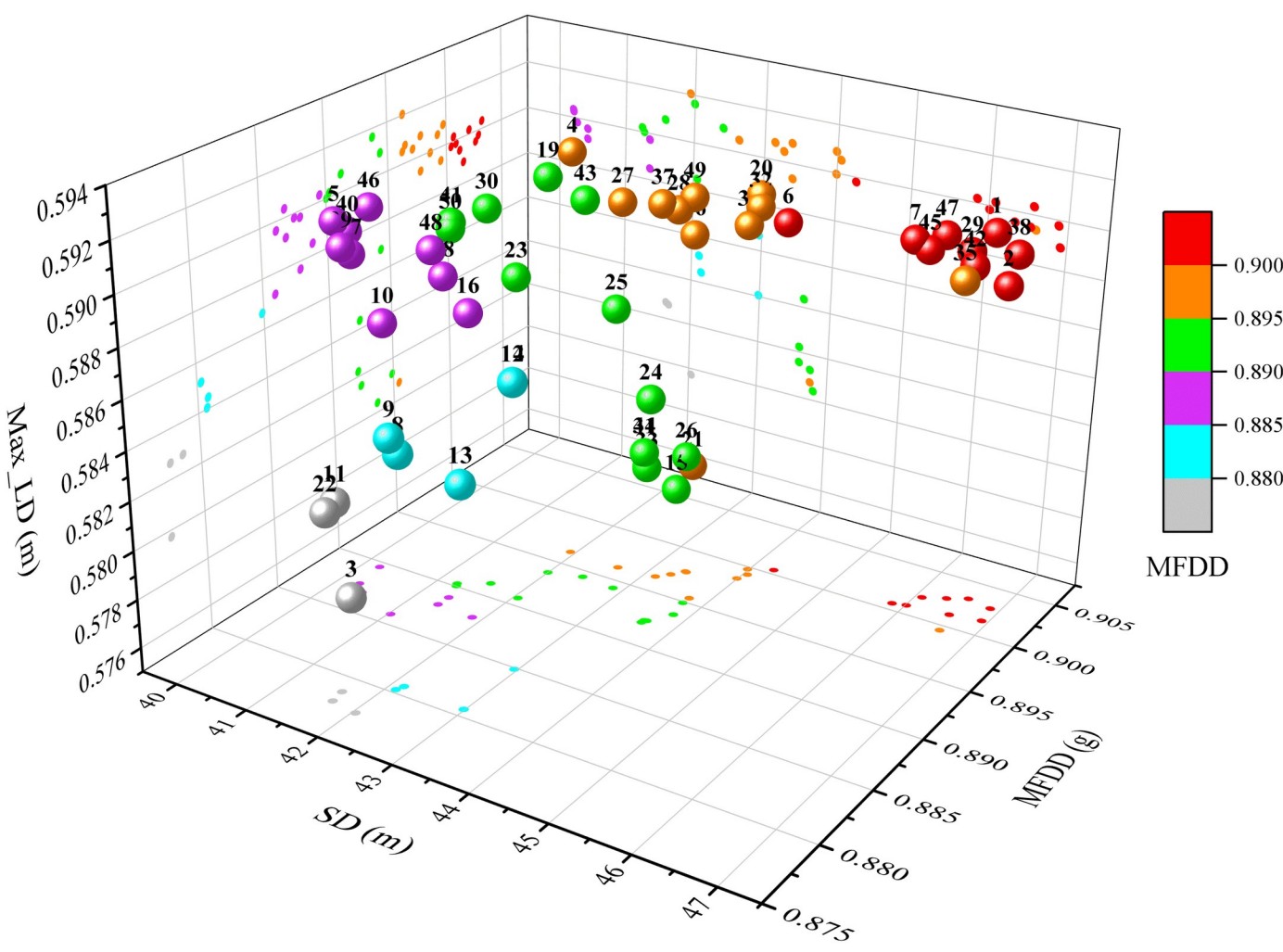

**Fig 12. Optimal results.**

- In the interval of 0.880 g ≤ MFDD < 0.885 g, the 8th individual was selected;

- In the interval of MFDD < 0.880 g, the 11st individual was selected.

The objective values and the non–dominated sorting results correspond to the selected individuals are shown in Table 9. Aiming to compare the differences among the individuals more clearly, they were represented by radar map again, and as shown in Fig 14.

As can be seen from the Fig 14:

- the CD and the SD of the 6th individual are disadvantage except for the relatively maximum MFDD;

- the LD of the 4th individual is larger, and the SD is not dominant compared with other individuals;

- although the 8th and the 11st individuals have minimum Max LDs, but their MFDDs are also small, and their SDs are also not dominant simultaneously.

Compared with the 30th and the 40th individuals, although the 30th individual has advantages in the MFDD and the CD, while it is inferior in the LD and the SD. In addition, the SD

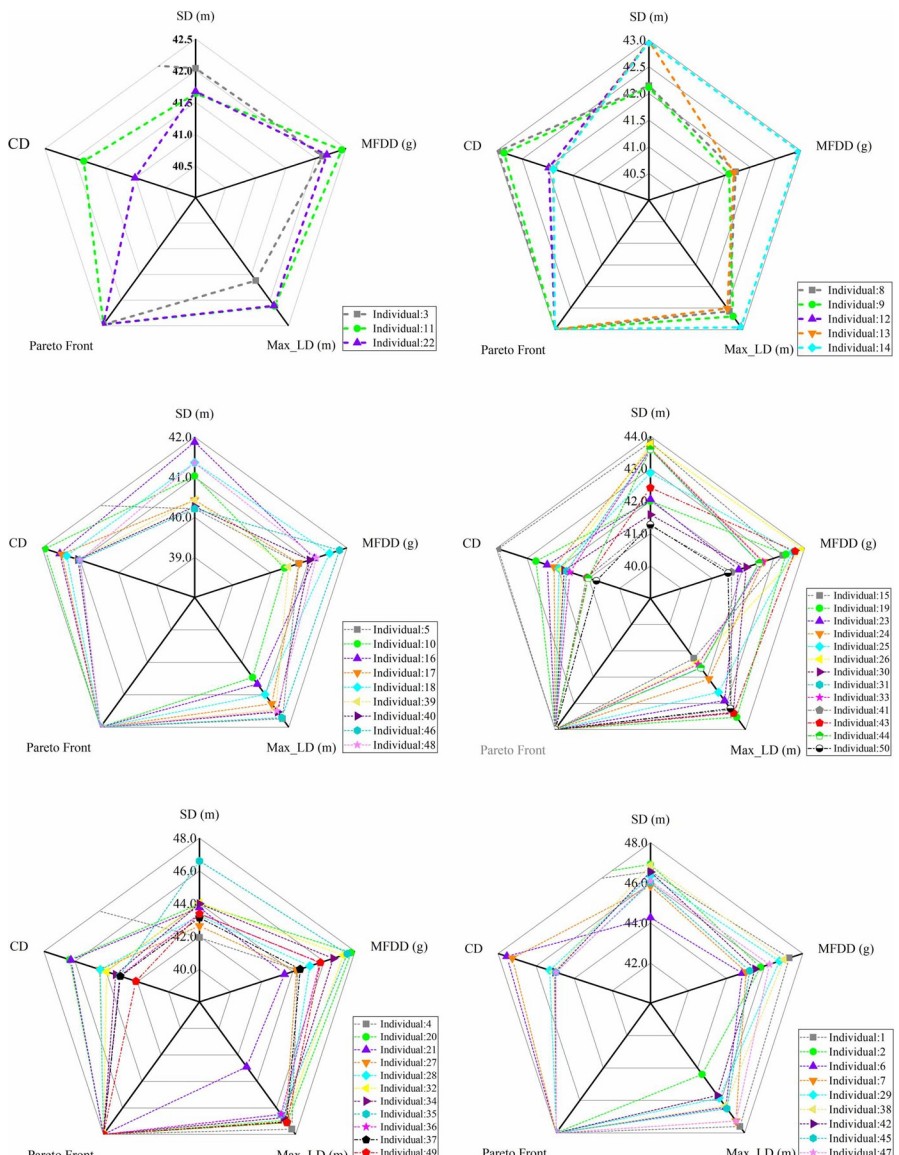

**Fig 13. Individuals under the different MFDD intervals.** (a) MFDD < 0.880 g; (b) 0.880 g ≤ MFDD < 0.885 g; (c) 0.885 g ≤ MFDD < 0.890 g; (d) 0.890 g ≤ MFDD < 0.895 g; (e) 0.895 g ≤ MFDD < 0.900 g; (f) MFDD ≥ 0.900 g.

**Table 9. Selected individuals.**

| Individuals | Objective values | | | Non–dominated sorting results | |
|---|---|---|---|---|---|
| | SD/m | MFDD/g | Max LD/m | Pareto front | CD |
| 4 | 41.935 | 0.896 | 0.592 | 1 | inf |
| 6 | 44.263 | 0.900 | 0.589 | 1 | 0.307 |
| 8 | 42.147 | 0.881 | 0.584 | 1 | 0.277 |
| 11 | 41.639 | 0.879 | 0.583 | 1 | 0.223 |
| 30 | 41.579 | 0.891 | 0.591 | 1 | 0.095 |
| 40 | 40.266 | 0.888 | 0.590 | 1 | 0.074 |

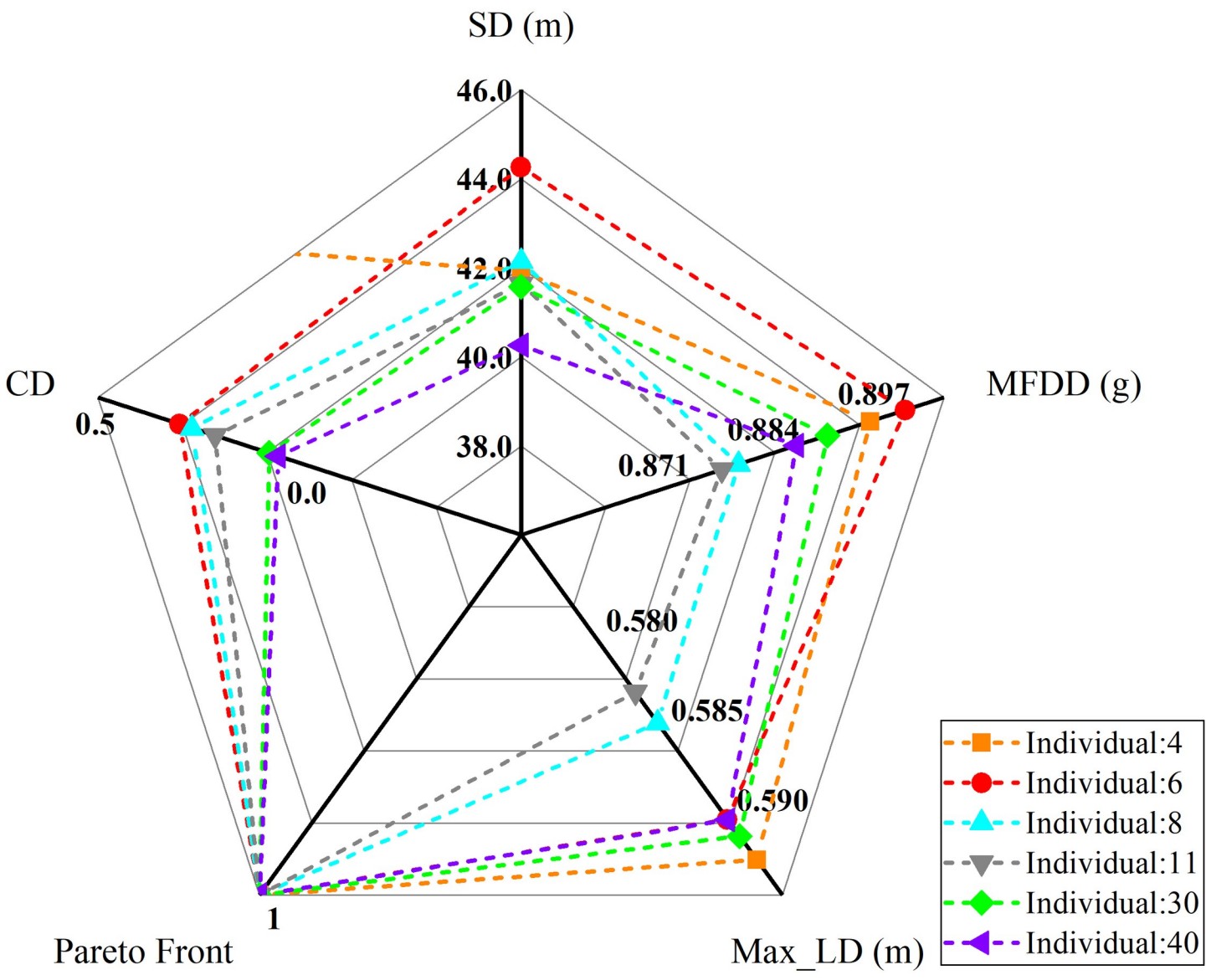

**Fig 14. Selected individuals.**

of the 40<sup>th</sup> individual has the most prominent advantage among the six selected individuals, and its MFDD and Max LD are relative equilibrium.

Hence, the 40<sup>th</sup> individual was selected as the final satisfactory solution, and the corresponding decision variable values were taken as the optimal results. The comparisons of the decision variables are listed in Table 10.

### 4.3 Simulation analyses

**4.3.1 Braking pressure control.** When the vehicle is stationary, a step signal of maximum braking pressure is applied to the EMB. Fig 15 shows the response curves of the braking pressure, the braking torque, the motor speed and the motor torque.

**Table 10. Comparison of the decision variables.**

| Parameters | Structural parameters | | | | | | Control parameters | | | | | |
|---|---|---|---|---|---|---|---|---|---|---|---|---|
| | $g$ | $L$ | $J$ | $\psi_m$ | $D$ | $p$ | $k_{p\_F}$ | $k_{i\_F}$ | $k_{d\_F}$ | $k_1$ | $k_2$ | $\alpha$ |
| Before optimization | 4 | 5 | 0.00113 | 0.007 | 0.0015 | 3 | 1500 | 0.15 | 0.1 | 6 | 8 | 0.1 |
| After optimization | 3.09 | 5.98 | 0.001 | 0.01 | 0.001 | 4 | 1545.54 | 0.1 | 0.05 | 7.83 | 8.86 | 0.12 |

As observed from Fig 15A and 15B, the braking pressure and the braking torque reach the maximum values of 120 kN and 16500 N·m at 0.5 s after optimization, and the EMB eliminates the brake clearance at 0.06 s. These two indexes were shortened by approximately 0.3 s and 0.04 s respectively.

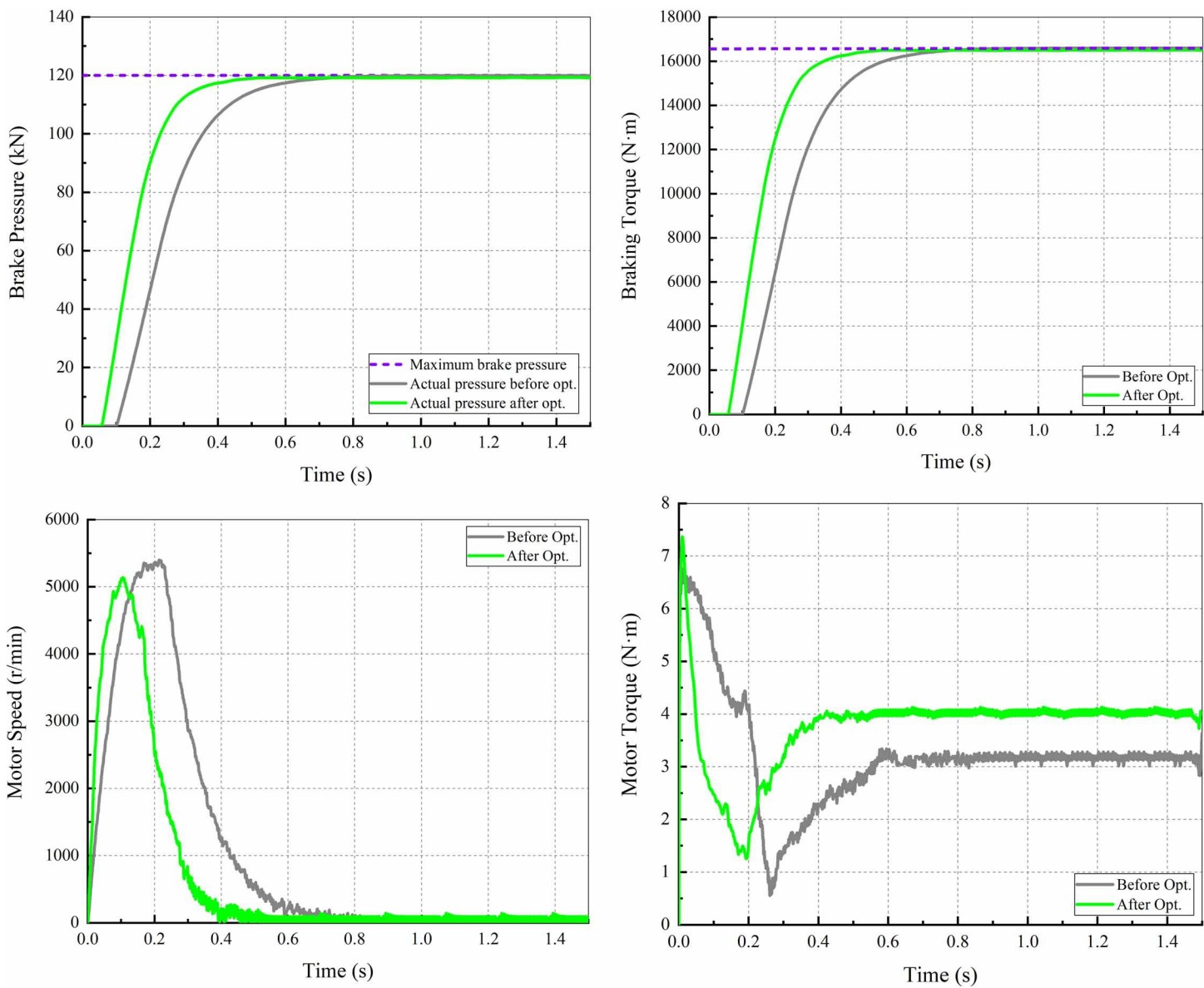

**Fig 15. Response curves of the EMB.** (a) Braking pressure; (b) Braking torque; (c) Motor speed; (d) Motor torque.

It can be seen from Fig 15C and 15D that the maximum motor speed is approximately 5000 r/min. In contrast, the motor responses under the optimal parameters are more positive, and the continuous motor torque is large. This means that the optimized motor parameters can match the smaller reduction mechanism, thus reducing the system inertia and overall size under the same EMB performance.

**4.3.2 ABS control.**    The vehicle is set to brake on a straight and dry road with an adhesion coefficient of 0.85 at an initial speed of 90km/h. The curves of the vehicle velocities, the wheel center velocities and the wheel slip ratios are shown in Fig 16. As observed from the figures that the ABS control prototype can keep the slip rates of each wheel near the ideal value (about 0.2) in the initial braking stage. When the vehicle velocity is less than 5 km/h at approximately 3 s, the ABS is not activated and the wheels are locked temporarily, which are consistent with the expectations.

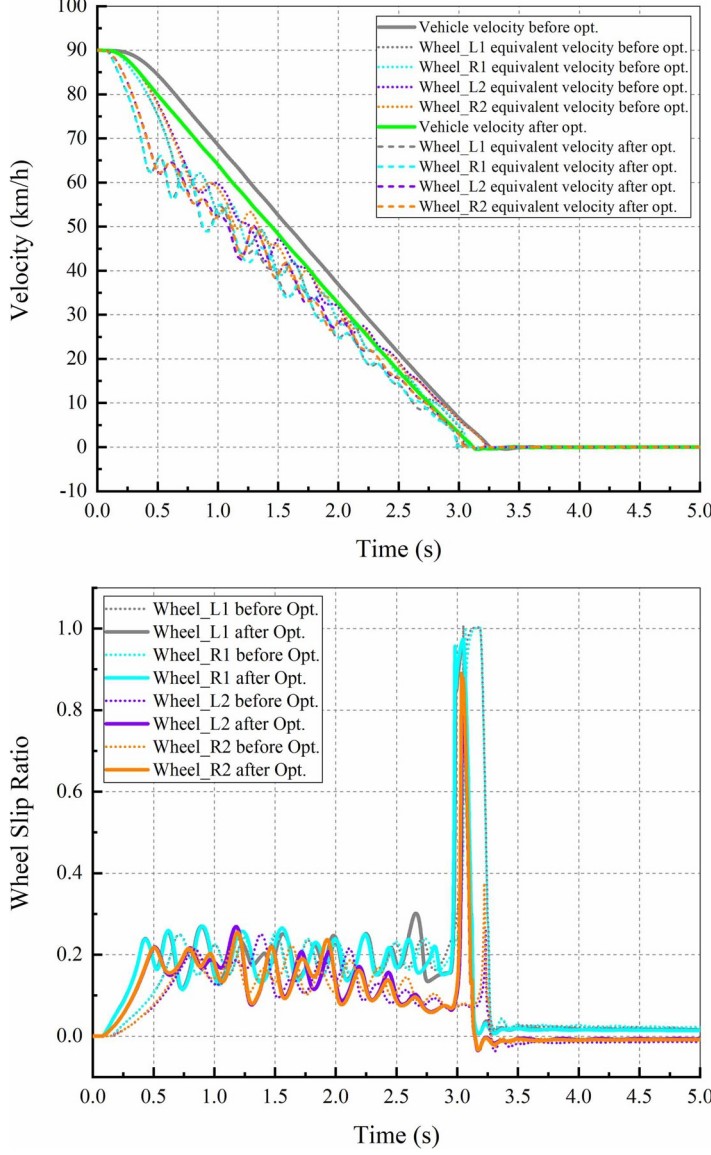

**Fig 16. Comparison of ABS control.** (a) Vehicle and equivalent wheel velocity; (b) Wheel slip ratios.

It should be noted that: from Eq (8), it can be seen that the adhesion coefficient between the tire and the ground is a crucial parameter in the ABS control law. In fact, the primary task of an ABS is to identify the adhesion condition between the tire and the ground. However, the main aim of this study is not to design an ABS control strategy. Therefore, this study simplifies the recognition algorithm of the adhesion coefficient, and sets the coefficient to a certain value ($\mu = 0.85$) directly according to the test conditions specified by the relevant standards and regulations [26–28].

**4.3.3 Braking performances.** During the braking process, the driver holds the steering wheel tightly and does not correct the vehicle direction. The braking performance is compared by the SD, the MFDD and the LD shown in Figs 17 and 18.

Fig 17 shows the longitudinal distance with time. It can be seen from the figure that the time from braking to stopping at the initial velocity of 90 km/h is approximately 3 s, and the SD after optimization is 40.26 m. These two indexes were shortened by approximately 0.22 s and 3.44 m respectively.

Fig 18 shows the deceleration with time. Under the action of ABS, the vehicle's maximum deceleration fluctuates around 0.9 g. After calculation, the optimized MFDD is 0.888 g and improved approximately 0.002 g than the result before optimization.

Fig 19 shows the LD with time and longitudinal distance. From the figure, the vehicle is shifted to the left in the forward when braking. Moreover, the Max LD before optimization is

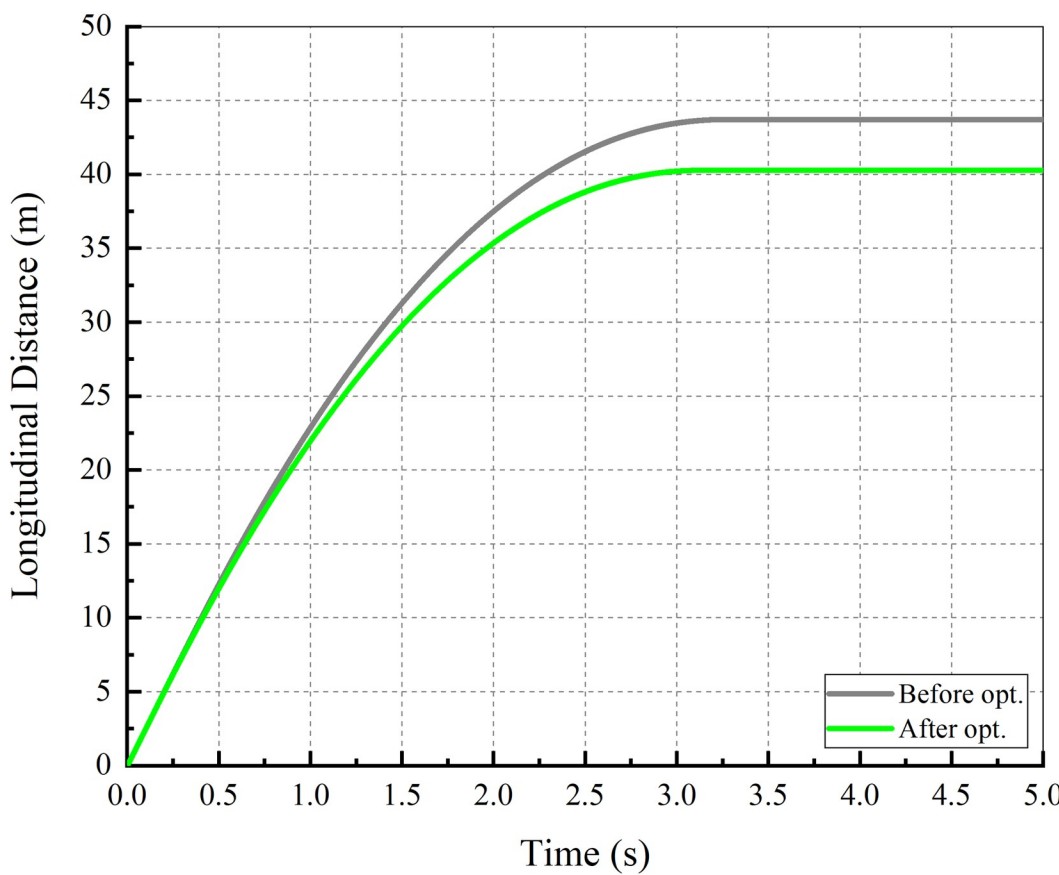

**Fig 17. Longitudinal distance–time.**

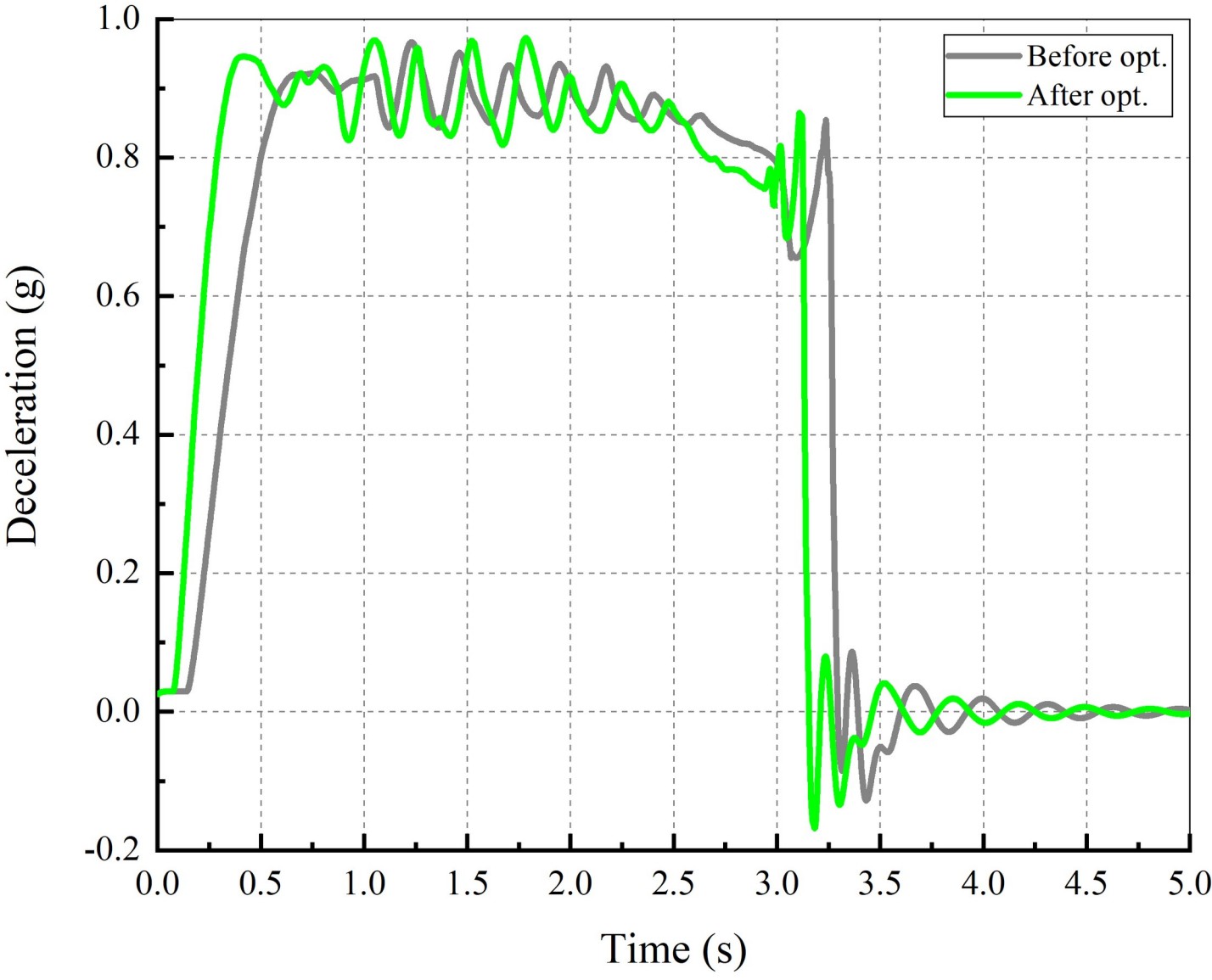

**Fig 18. Deceleration–time.**

approximately 0.627 m, and the result is approximately 0.59 m after optimization. Hence, the braking stability is improved.

The comparisons of the indicators are shown in Table 11. From the table, the response time of maximum braking pressure and the SD are obvious improved after optimization. Specifically, the response time is shortened by approximately 37.5%, the SD is shortened by approximately 7.87%, and other indexes are improved.

## 5. Conclusions

The novelty of this study is to provide a new idea for the parameters optimization of vehicle braking systems. In this way, the influence of the load transfer, the braking distribution, the ABS effects and the parameters coupling on the vehicle's braking performance can be reduced, and the vehicle performance can be optimized. This idea can be extended to the

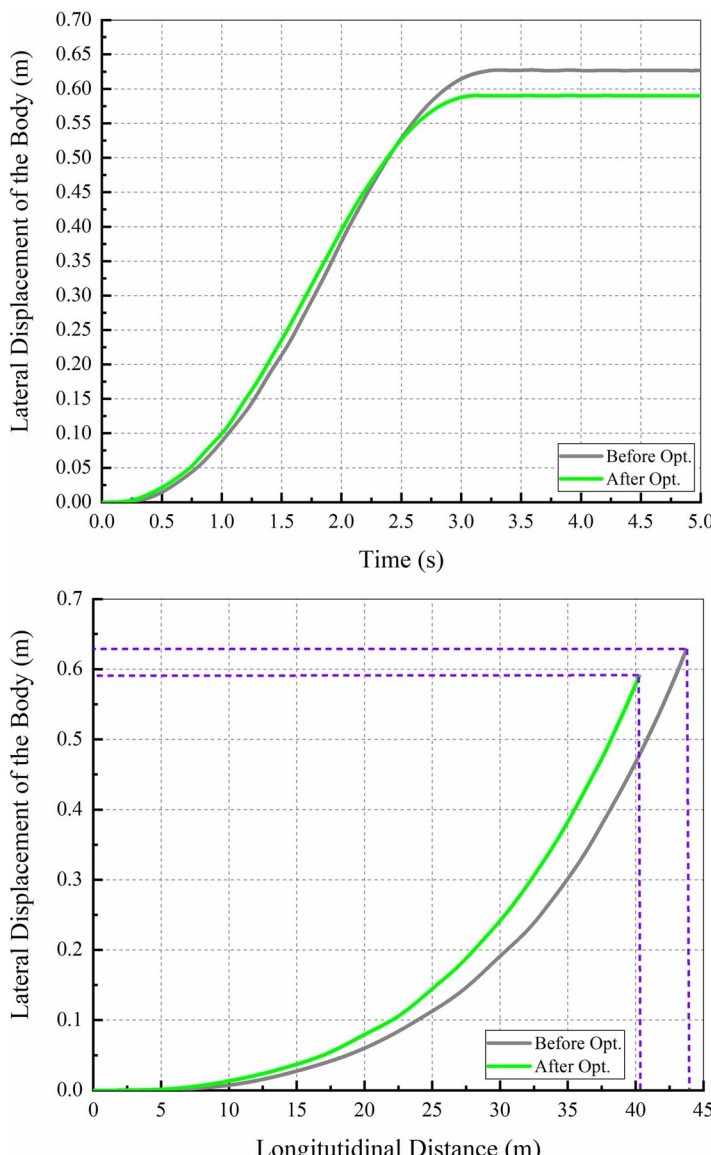

**Fig 19. LD with time and longitudinal distance.** (a) LD–Time; (b) LD–Longitudinal distance.

components design and systems optimization of vehicle subsystems, not limited to the braking system.

The following work was conducted: First, on the basis of our previous study, an upgraded EMB scheme was proposed, and the corresponding system model and the control prototype

**Table 11. Comparison of the indicators.**

| Indicators | Before optimization | After optimization | Effects |
|---|---|---|---|
| Maximum braking pressure response time | 0.8 s | 0.5 s | shortened 0.3 s |
| SD of 90 km/h–0 | 43.7 m | 40.26 m | shortened 3.44 m |
| MFDD | 0.886 g | 0.888 g | improved 0.002 g |
| Max LD | 0.627 m | 0.59 m | shortened 0.037 m |

were established. Then, from the perspective of vehicle's braking performance, the crucial structural and control parameters of the EMB were taken as decision variables, and the ranges of the parameters, the maximum braking torque provided by the ground and the relevant standards were set as constraints to establish the optimal mathematical model, and the co–simulation platform of MATLAB/Simulink and TruckSim was constructed. Finally, the EMB parameters were optimized based on the NSGA–II and the results were discussed and simulated based on the established platform.

The results show that:

- After optimization, the step response time of maximum braking pressure is shortened by approximately 0.3 s;

- The ABS control prototype can keep the slip ratio near the ideal value at the initial braking stage. Under the action of the ABS, the optimal MFDD is improved by approximately 0.002 g;

- After the optimization, the SD of 90 km/h–0 and the Max LD are shorten by approximately 3.44 m and 0.037 m.

In general, the EMB response time and the SD are improved obviously, and other indexes and the vehicle's braking performance are all improved after optimization.

Future work can address fabricating an EMB functional prototype for field and road tests to validate the optimization further. And a coordinated control and optimal distribution strategy of the EMB braking pressure maybe a meaningful study.

## Supporting information

**S1 Appendix. Orthogonal tests results.**
(DOCX)

**S2 Appendix. Range analyses.**
(DOCX)

## Author Contributions

**Data curation:** Xuan Qin.

**Formal analysis:** Tong Wu, Xuan Qin.

**Funding acquisition:** Jing Li.

**Investigation:** Xuan Qin.

**Methodology:** Tong Wu.

**Project administration:** Jing Li.

**Software:** Tong Wu.

**Validation:** Xuan Qin.

**Writing – original draft:** Tong Wu.

**Writing – review & editing:** Tong Wu.

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
