## [Decision Letter · Decision Letter 0]

19 Feb 2021

PONE-D-20-39894

Braking performance oriented multi–objective optimal design of electro–mechanical brake parameters

PLOS ONE

Dear Dr. Tong Wu,

Thank you for submitting your manuscript to PLOS ONE. After careful consideration, we feel that it has merit but does not fully meet PLOS ONE’s publication criteria as it currently stands. Therefore, we invite you to submit a revised version of the manuscript that addresses the points raised during the review process.

We look forward to receiving your revised manuscript.

Kind regards,

Jing Zhao, Ph.D.

Academic Editor

PLOS ONE

Journal Requirements:

2. Please upload a new copy of Figure 4, 5 and 10 as the detail is not clear. Please follow the link for more information: https://blogs.plos.org/plos/2019/06/looking-good-tips-for-creating-your-plos-figures-graphics/" https://blogs.plos.org/plos/2019/06/looking-good-tips-for-creating-your-plos-figures-graphics/

Reviewers' comments:

Reviewer's Responses to Questions

**Comments to the Author**

1. Is the manuscript technically sound, and do the data support the conclusions?

Reviewer #1: Yes

Reviewer #2: Partly

Reviewer #3: Yes

2. Has the statistical analysis been performed appropriately and rigorously? 

Reviewer #1: Yes

Reviewer #2: N/A

Reviewer #3: N/A

3. Have the authors made all data underlying the findings in their manuscript fully available?

Reviewer #1: Yes

Reviewer #2: Yes

Reviewer #3: Yes

4. Is the manuscript presented in an intelligible fashion and written in standard English?

Reviewer #1: Yes

Reviewer #2: Yes

Reviewer #3: Yes

5. Review Comments to the Author

Reviewer #1: The manuscript is technically sound, and the data can support the conclusions. But there are the following problems:

1. The structure and working principle of EMB should be detailed, especially the transfer relationship.

2. Explain the Figure 2 in detail, for example, why the unit of measurement point is MPa and the unit of ordinate is kN.

3. The description for ABS control should be detailed.

4. The meaning of some variables is not specific, for example, “kp, ki and kd are the parameters”.

5．The meaning of some letters is missing，such as the letters in table2.

6. There are two variables in table 1 and table 2, both of which are signed M.

Reviewer #2: 1. The authors claim that the EMB is better than a pneumatic brake. How about comparison with an electro-pneumatic brake? Further, what can the authors comment on the cost of an EMB vis-à-vis a pneumatic brake? Further, what ails the adoption of EMB?

2. The authors cite their earlier study (reference [3]) for details on the modeling process. Since this study seems to build upon their earlier study, what are the specific contributions with regards to modeling?

3. How is Eq. (2) obtained? The authors cite reference [19], but can they provide a physics-based explanation?

4. How are the adhesion coefficient and optimal slip ratio obtained? The knowledge of these parameters is important for practical implementation.

5. How do the authors address the system delays and nonlinearities?

6. How were the values of the EMB system model parameters (presented in section 2.4) obtained?

7. How did the authors get Eq. (13)? Further, it would be better to use appropriate units such that constants such as ‘3.6’ and ’25.92’ are avoided.

8. What are the original contributions with respect to the optimization process?

9. How would the ABS control discussed in section 4.3.2 work in the presence of unknown tyre-road interface conditions?

10. The quality of the figures/plots requires improvement.

Reviewer #3: Comments and Suggestions for Authors

The manuscript entitled “Braking performance oriented multi–objective optimal design of electro–mechanical.”

This study is to provide a new idea for the parameters optimization of vehicle braking systems. In this way, the influence of the load transfer, the braking distribution, the ABS effects and the parameters coupling on the vehicle’s braking performance can be reduced, and the vehicle performance can be optimized. Therefore, the following work was conducted. the presented results of this work could encourage the overall uses of multivariate methods in these fields. All presented results are well discussed. It is also easy to understand. also, the paper structure is suitable and well written. In addition, the figures and table are clear and useful. Also, this paper is interesting and well written.

I encourage the journal to accept it with minor corrections.

1. The authors should indicate the possible impact of their work. Also need to add the novelty and main contributions clearly. I encourage the authors to add problem statement in the abstract or introduction.

2. The authors should discuss the problem statement and the novelty clearly in the introduction part.

3. The authors should be revised the conclusion with proper key results.

4. Figures 4, 5, 9 and 10 are not clear should be enhance.

5. I suggest adding your main contributions comparing with other researchers clearly (what is the difference with other researchers) what is the new in your study... hope you could add it in your paper.

6. I suggest adding future recommendations to the next researchers after the conclusion part.

6. PLOS authors have the option to publish the peer review history of their article (what does this mean?). If published, this will include your full peer review and any attached files.

Reviewer #1: No

Reviewer #2: No

Reviewer #3: No

---

## [Author Response · Author response to Decision Letter 0]

26 Mar 2021

Dear Editor and Reviewers:

 Thank you for your letter and for the reviewers’ comments concerning our manuscript entitled “Braking performance oriented multi–objective optimal design of electro–mechanical brake parameters” (Manuscript Number: PONE-D-20-39894). Those comments are all valuable and very helpful for revising and improving our manuscript, as well as the important guiding significance to our researches. We have studied comments carefully and have made correction which we hope meet with approval. 

 We keep the original reviewers’ comments in green

 Revised parts in the manuscript are marked in red

 Added parts in the manuscript are marked in blue

 Our responses and the unchanged parts in the original manuscript are marked in black

 The main corrections in the paper and the responds to the reviewer’s comments are as flowing:

Reviewer #1: The manuscript is technically sound, and the data can support the conclusions. But there are the following problems:

1. The structure and working principle of EMB should be detailed, especially the transfer relationship.

Response：Thank you very much for your work and comments. Therefore, we revised the Section 2.1: firstly, the structural advantages of the pneumatic–disc–brake are pointed out, then the structure characteristics, working principle and transfer relationship of the EMB are introduced. Finally, the improved EMB is compared with the previous EMB, as follows:

 “Note that the axial thrust output from the push rod (6) is amplified and so the size of the air chamber could be reduced due to the arm (5). Furthermore, the caliper already existed a mechanism to adjust the brake clearance. Therefore, the caliper is still retained in the improved EMB.

 Particularly, to retain the caliper, the brake chamber is replaced by the powerplant to push the mechanisms inside the caliper that generates the braking pressure [3]. While, in the proposed EMB, the pressure pushing the push rod (6) is generated by a motor (15), a reduction mechanism (14) and a screw mechanism (12), and the power transmission path of the EMB in the powerplant is shown with blue arrows in Fig 1b. During the working of the system, the torque output from the motor is amplified by the reduction mechanism (14) and transformed by the screw (12) into an axial thrust force on the push rod (6) and act on the arm (5) in the caliper. The transmission path of the axial thrust force in the caliper is exactly the same as the pneumatic–disc–brake shown in Fig 1a. Furthermore, the pressure sensor (11) measures the axial thrust and the linear sensor (13) measures the distance axially traveled by the screw.

 Compared with the previous scheme, the improved EMB reduces the requirements of motor performances attribute to the reduction mechanism.”

2. Explain the Figure 2 in detail, for example, why the unit of measurement point is MPa and the unit of ordinate is kN.

Response：The caliper model derived based on dynamic theories is complex and contains many unknown parameters, and its accuracy needs to be verified further. Therefore, the curve shown in Fig 2 is used as the caliper model, which describes the relationship between the axial displacement of the push rod (6) and the normal pressure on the disc (2). Where, series points in Fig 2 are measured and provided by the caliper manufacturer based on the pneumatic–disc–brake, so the unit of the points are MPa.

 Since the caliper is retained and the chamber in the pneumatic brake is replaced by the powerplant in the proposed EMB, so the unit of pressure is converted from MPa to kN, and series points are fitted to the curve shown in Fig 2.

 We would like to thank the reviewer’s question, which makes us realize that the expression may confuse the readers. Therefore, we divide Fig 2 into two parts: Fig 2a is the original data provided by the caliper manufacturer, and Fig 2b is the caliper model after unit conversion and fitting. To this, we add a note after the 3rd paragraph of Section 2.2:

 “Fig 2 shows the caliper model, where Fig 2a is the pneumatic–disc–brake based test data using a 19.5 inch chamber provided by the manufacturer. Fig 2b is the caliper model available for the proposed EMB after unit conversion (from MPa to kN) and fitting based on Fig 2a.”

3. The description for ABS control should be detailed.

Response：Thanks to the reviewer’s suggestion. For this reason, we add a description about the ABS working process and its principle in Section 2.3.3, and as follows:

 “The ABS is to identify the states of the wheels, to avoid the wheels locking by adjusting the wheel braking pressure in proper time, and to ensure braking stability of the vehicle. In the traditional braking systems, the ABS is realized by adjusting the pressure in the braking pipe through a solenoid valve. While, due to the braking pressure of each wheel is controlled independently, the ABS based on EMBs is more flexible.

 A crucial index to measure the wheel states is the slip ratio λ, which indicates the proportion of wheel slip in pure rolling when braking. 

 The slip ratio λ determines the longitudinal force Fxb and the lateral force Fy acting on the wheels. Fig 5 shows the relationships between the longitudinal force Fxb and the lateral force Fy with the slip ratio λ. It can be seen from the figure that the lateral force Fy decreases with the slip ratio λ. When the slip ratio λ reaches 100%, the lateral force Fy is 0, and the vehicle will lose the steering ability or fall into the risk of tail flick. At the same time, the longitudinal force Fxb increases rapidly with the slip ratio λ at the beginning, when the slip ratio λ reaches to a certain range, the longitudinal force Fxb gradually decreases with the slip ratio λ.

 To ensure the wheels are subjected to a greater braking force and a certain lateral force which significant to the vehicle’s lateral stability, the ABS should keep the slip ratio λ of each wheel near the optimal slip ratio λ*. The test data show that the optimal slip ratio λ* of pneumatic tire is about 15 % ~ 20 % when driving on a rigid road [19]. In this paper, the optimal slip ratio λ* is set to 20 %.”

4. The meaning of some variables is not specific, for example, “kp, ki and kd are the parameters”.

Response：Thanks to the reviewer’s reminder. While, the kp, the ki and the kd are the parameters of the PID algorithm. In order to avoid ambiguity, we modify the text as “kp, ki and kd are the PID parameters” in the manuscript.

 Furthermore, we found out that there is similar problem in Section 2.3.3, text “k1, k2 and α are coefficients”. We also revised the text as “k1, k2 and α are the SMC parameters”

5．The meaning of some letters is missing，such as the letters in table 2.

Response：Thanks to the reviewer, under your reminder we found that the meanings of the letters m and a in Table 2 are missing. So we supplemented them below table 2:

 “Where, m is the full load mass of the vehicle, and a is the distance between the centroid and the front axle center, and the rest symbols have the same meaning as before.”

6. There are two variables in table 1 and table 2, both of which are signed M.

Response：Thanks to the reviewer, we have noticed the phenomenon of repeated variable name. In fact, the variable “M ” in Table 1 means the mutual inductance of the phase, while the variable “M ” in Table 2 means the full load mass of the vehicle. To avoid ambiguity, we replace the variable “M ” in Table 2 with “m ” and modify it wherever it appears.

 In addition, we also note that the variable “α” in Table 1 is named repeatedly with the variable in Table 3. Where, the variable “α” in Table 1 means helix angle of the screw, while the variable “α” in Table 3 is an ABS control parameter. To avoid ambiguity, we replace the variable “α” in Table 1 with “ζ” and modify it wherever it appears.

Reviewer #2: 

1. The authors claim that the EMB is better than a pneumatic brake. How about comparison with an electro-pneumatic brake? Further, what can the authors comment on the cost of an EMB vis-à-vis a pneumatic brake? Further, what ails the adoption of EMB?

Response：Thank you very much for your valuable comments. For question 1 raised by the reviewer, we consider this way:

 Compared with the pneumatic brake, the electro-pneumatic brake can directly control the proportional valve, relay valve and bridge valve by electronic signal to establish the braking force, thus eliminating the shortcomings such as slow response and poor comfort of the conventional pneumatic brake. Moreover, the electro-pneumatic brake also improving the brake consistency between towing vehicle and trailer, integrating ABS, ESC, TCS and other functions, so the braking safety and comfort is improved. However, the electro-pneumatic brake still needs air tanks, dryer, pneumatic pipelines and other components, and there are still exhaust noise and other problems during braking. While, these phenomena can be solved in the EMB (mentioned in the 2nd paragraph of Section 1).

 In terms of cost, because the electro-pneumatic brake is the current mainstream scheme of commercial vehicle, its technology is mature and performance is stable. In the short term, the cost of EMB is indeed higher than the electro-pneumatic brake. While, the EMB has the advantages of simple structure, flexible control, high integration and convenient matching, and can meet the braking requirements of autonomous driving and intelligent connected vehicles.

 It is undeniable that the reliability of EMB still needs to be further improved, and the reliability is the crucial factor restricting the development of EMB even brake-by-wire (BBW) system. Because the EMB breaks the mechanical connection between the brake pedal and the actuator, the reliability of the system will be further tested. To design the EMB software and hardware architectures with redundancies, and establish the emergency strategy when the system fails are the urgent problems to be solved in the stage of EMB development and application.

2. The authors cite their earlier study (reference [3]) for details on the modeling process. Since this study seems to build upon their earlier study, what are the specific contributions with regards to modeling?

Response：We consider the question from two aspects:

 Firstly, in terms of structure, the EMB proposed in this paper is similar to the EMB mentioned in reference [3]. The difference is that the proposed EMB adds a reduction mechanism between the motor and the screw (as shown in Fig 1b). The advantage of this change is the performance requirements, the size and the cost of the motor can be reduced. Therefore, compared with the EMB system model in reference [3], this study adds the reduction ratio g into the EMB model.

 In addition, in the previous study, we found that excessive brake clearance will affect the response performance of the EMB. Therefore, in this study, we contacted the caliper manufacturer to adjust the brake clearance and test the caliper characteristics again. Because the caliper already has a mechanism to adjust the clearance and the structure of the caliper remains unchanged during two tests, so the trend of the curve (as shown in Fig 2) in this paper is almost unchanged and the position of the curve moves to the left compared with that in reference [3].

3. How is Eq. (2) obtained? The authors cite reference [19], but can they provide a physics-based explanation?

Response：Eq. (2) is the basic principle of vehicle dynamics. Due to the limited space, we did not show the complete derivation process in the manuscript and in here. The physical explanation and derivation of Eq. (2) 

please refer to the attachment “Response to reviewers” for details.

4. How are the adhesion coefficient and optimal slip ratio obtained? The knowledge of these parameters is important for practical implementation.

Response：The questions raised by the reviewer are very meaningful and crucial. We response from two aspects：

 1) In fact, adhesion coefficient is a crucial parameter in ABS control. In an actual ABS control system, the adhesion coefficient is identified by an certain algorithm. It is a very important task for an ABS strategy to identify the adhesion coefficient, and the result will directly affect the control quality of the ABS. 

 However, the main idea of this manuscript is not to design an ABS control strategy, but to propose an optimization idea which considers the brake structure and braking control parameters at the same time, so as to reduce the impacts such as load transfers, braking distributions and parameters coupling on vehicle’s braking performance. 

 Therefore, we set the adhesion coefficient as 0.85 (horizontal dry asphalt pavement) according to the test conditions of vehicle braking performance stipulated in the ECE R13, the GB 7258-2017, the GB 12676-2014 and other relevant standards.

 To this, we add a paragraph in Section 4.3.2 of the manuscript for supplementary explanation, and as follows: 

 “It should be noted that: from equation (8), it can be seen that the adhesion coefficient between the tire and the ground is a crucial parameter in the ABS control law. In fact, the primary task of an ABS is to identify the adhesion condition between the tire and the ground. However, the main aim of this study is not to design an ABS control strategy. Therefore, this study simplifies the recognition algorithm of the adhesion coefficient, and sets the coefficient to a certain value (μ = 0.85) directly according to the test conditions specified by the relevant standards and regulations [26–28].”

 2）Furthermore, we added a description of the ABS working principle and the determination of optimal slip ratio in Section 2.3.3, and as follows: 

 “ The ABS is to identify the states of the wheels, to avoid the wheels locking by adjusting the wheel braking pressure in proper time, and to ensure braking stability of the vehicle. In the traditional braking systems, the ABS is realized by adjusting the pressure in the braking pipe through a solenoid valve. While, due to the braking pressure of each wheel is controlled independently, the ABS based on EMBs is more flexible.

A crucial index to measure the wheel states is the slip ratio λ, which indicates the proportion of wheel slip in pure rolling when braking. 

 The slip ratio λ determines the longitudinal force Fxb and the lateral force Fy acting on the wheels. Fig 5 shows the relationships between the longitudinal force Fxb and the lateral force Fy with the slip ratio λ. It can be seen from the figure that the lateral force Fy decreases with the slip ratio λ. When the slip ratio λ reaches 100%, the lateral force Fy is 0, and the vehicle will lose the steering ability or fall into the risk of tail flick. At the same time, the longitudinal force Fxb increases rapidly with the slip ratio λ at the beginning, when the slip ratio λ reaches to a certain range, the longitudinal force Fxb gradually decreases with the slip ratio λ.

 To ensure the wheels are subjected to a greater braking force and a certain lateral force which significant to the vehicle’s lateral stability, the ABS should keep the slip ratio λ of each wheel near the optimal slip ratio λ*. The test data show that the optimal slip ratio λ* of pneumatic tire is about 15 % ~ 20 % when driving on a rigid road [19]. In this paper, the optimal slip ratio λ* is set to 20 %.”

5. How do the authors address the system delays and nonlinearities?

Response：1) The EMB system delays are mainly caused by the delay and inertia links. There are many probabilities for the delay links: In terms of structure, they are mainly caused by the clearances in the reduction mechanism, screw mechanism and brake; in terms of control system, there are delays in the three-phase circuit components and the signal measurements. Furthermore, the inertia links are mainly caused by the system inertia and motor response performances.

 Based on the reasonable matching, the EMB parameters are further optimized to minimize the system delays caused by the brake clearance, system inertia, motor response performances and other factors.

 Here, we would like to thank the reviewer’s comments, which reminds us: For the system delays caused by the clearances in the mechanisms, the three-phase circuit components, the signal measurements and other factors, are worth further study to improve the EMB synthesis.

 2) The nonlinear of the system is mainly caused by the friction in the screw mechanism. For this reason, this study first introduce the friction model of the mechanism, and then determine the friction torque during the mechanism movement based on the friction model. Finally, a compensation current is determined to overcome the nonlinear friction in the screw mechanism by making the motor output an additional torque. Please refer to Section 2.3.4 for details.

 Moreover, the working of ABS has strong nonlinear characteristics. Therefore, we deduce the ABS control law based on the sliding mode control (SMC) algorithm. The SMC is an appropriate way to solve nonlinear systems, and it demonstrates the characteristics of fast response, insensitive to parameter changes and external disturbances.

6. How were the values of the EMB system model parameters (presented in section 2.4) obtained ?

Response: The parameters of the EMB system model described in Section 2.4 include three aspects: the EMB system model parameters (Table 1), the vehicle related parameters (Table 2) and the initial parameters of the control prototype (Table 3). Among them, the vehicle related parameters shown in Table 2 are provided by the vehicle manufacturer, the caliper force amplification factor (k) in Table 1 is provided by the caliper manufacturer, and the initial parameters of the control prototype shown in Table 3 are obtained by our tuning to facilitate the multi-objective optimization design.

 Furthermore, to determine the initial value of the optimization variables, the motor related parameters (R, L, M, p, ψm, D, J), the screw related parameters (dm, l, ƞ, α, ρ) and the reduction ratio g shown in Table 1 are obtained by our preliminary matching. The matching idea is: First, according to the EMB performance requirements, the caliper model and the vehicle dynamics, the required motor torque, speed and power are preliminarily obtained. Next, the benchmark motor which meeting the requirements is found by changing the reduction ratio and the screw lead range, and its parameters (R, L, M, p, ψm, D, J) are taken as the initial references. Then, the maximum axial thrust and the moving velocity of the screw can be calculated according to the motor parameters and the force analysis of the screw. Finally, the nominal diameter dm and the lead range l of the screw are determined based on the wear resistance principle, and then the helix angle α and the equivalent friction angle ρ are calculated. 

 Because the process of parameter matching is lengthy and not the main work of this paper, so we did not include it in the manuscript.

7. How did the authors get Eq. (13)? Further, it would be better to use appropriate units such that constants such as ‘3.6’ and ‘25.92’ are avoided.

Response: Eq. (13) is the formula of the stopping distance. Due to the limited space, we did not show the complete derivation process in the manuscript and in here. Please refer to the attachment “Response to reviewers” for details.

 The reviewer's suggestion is very reasonable. We have revised Eq. (13) to avoid constants such as “3.6” and “25.92”.

8. What are the original contributions with respect to the optimization process?

Response: The original contribution of the optimization is to provide a new idea for the parameters optimization of vehicle braking systems. Specifically, the first is to consider both the structural and the control parameters when selecting the decision variables; the second is to set the optimal objectives from the perspective of vehicle braking performance, which is no longer limited to the brake. In this way, the influence of the load transfer, the braking distribution and the parameters coupling can be decreased.

 The main contributions are described in paragraph 7 of Section 1 in the revised manuscript, and as follows:

 “Compared with other studies, the main contribution of this study is providing a new idea for the parameters optimization of vehicle braking systems. Specifically, the first is to consider both the structural and the control parameters when selecting the decision variables; the second is to set the optimal objectives from the perspective of vehicle braking performance, which is no longer limited to the brake responses. In these ways, the influences of the load transfers, the braking distribution and the parameters coupling can be decreased.”

9. How would the ABS control discussed in section 4.3.2 work in the presence of unknown tyre-road interface conditions?

Response: The reviewer's concern is necessary. In fact, the adhesion coefficient μ is a crucial parameter in the ABS control law derived in Section 2.3.3. The first task of the ABS is to identify the adhesion conditions between the tire and the road through algorithm.

 However, the main aim of this study is not to design an ABS control strategy. Therefore, this study simplifies the recognition algorithm of the adhesion coefficient, and sets the coefficient to a certain value (μ = 0.85) directly according to the ECE R13, the GB 7258-2017 and the GB 12676-2014.

 For this, we have made a supplementary explanation in Section 4.3.2, and as follows:

 “It should be noted that: from equation (8), it can be seen that the adhesion coefficient between the tire and the ground is a crucial parameter in the ABS control law. In fact, the primary task of an ABS is to identify the adhesion condition between the tire and the ground. However, the main aim of this study is not to design an ABS control strategy. Therefore, this study simplifies the recognition algorithm of the adhesion coefficient, and sets the coefficient to a certain value (μ = 0.85) directly according to the test conditions specified by the relevant standards and regulations [26–28].”

10. The quality of the figures/plots requires improvement.

Response: Thanks to the reviewer’s suggestion, we are also aware that the Fig 4, 5, 9 and 10 in the original manuscript are not clarity, and have improved the relevant figures again.

Reviewer #3: Comments and Suggestions for Authors

The manuscript entitled “Braking performance oriented multi–objective optimal design of electro–mechanical.” This study is to provide a new idea for the parameters optimization of vehicle braking systems. In this way, the influence of the load transfer, the braking distribution, the ABS effects and the parameters coupling on the vehicle’s braking performance can be reduced, and the vehicle performance can be optimized. Therefore, the following work was conducted. the presented results of this work could encourage the overall uses of multivariate methods in these fields. All presented results are well discussed. It is also easy to understand. also, the paper structure is suitable and well written. In addition, the figures and table are clear and useful. Also, this paper is interesting and well written. I encourage the journal to accept it with minor corrections.

1. The authors should indicate the possible impact of their work. Also need to add the novelty and main contributions clearly. I encourage the authors to add problem statement in the abstract or introduction.

Response：Thank you very much for your work and comments, we have made the following modifications to the manuscript:

 1) The main contribution is introduced, please refer to the paragraph 7 of Section 1 in details:

 “Compared with other studies, the main contribution of this study is providing a new idea for the parameters optimization of vehicle braking systems. Specifically, the first is to consider both the structural and the control parameters when selecting the decision variables; the second is to set the optimal objectives from the perspective of vehicle braking performance, which is no longer limited to the brake responses. In these ways, the influences of the load transfers, the braking distribution and the parameters coupling can be decreased.”

 2) The possible impact of our work is included in paragraph 1 of Section 5:

 “The novelty of this study is to provide a new idea for the parameters optimization of vehicle braking systems. … This idea can be extended to the components design and systems optimization of vehicle subsystems, not limited to the braking system. ”

 3) The abstract is revised and add a problem statement, and as follows:

 “… Meanwhile, many scholars have dedicated to the research on the parameters optimization of braking systems. While, most of the studies focus on reducing the brake size and weight, improving the brake responses by optimizing the parameters, almost not involving the braking performance, and the optimization variables are relatively single. On these foundations, a multi–objective optimal design of EMB parameters is proposed to enhance the vehicle’s braking performance. …”

2. The authors should discuss the problem statement and the novelty clearly in the introduction part.

Response：Thanks to the reviewer’s suggestion, we have made the following modifications to the manuscript:

 1) Revised the paragraphs 4 and 5 of Section 1. Where, the paragraph 4 is a summary of the previous studies; the paragraph 5 is a statement of the problem, and as follows: 

 “From the analyses, the optimal objectives in the [5–13] are the structural parameters, and the objectives in the [14–18] are the control parameters. Except for the [14] and the [17], the objectives in other studies are limited to improving the brake responses and reducing its size and weight.

 Actually, the performance of a brake is influenced by both the structural and the control parameters. Furthermore, the vehicle’s braking performance is not entirely determined by the brake response, but also impacted by such factors as the load transfer, the braking distribution and the control effect. Hence, the ultimate goal of the optimization should be improve the vehicle’s braking performance and not just the brake responses.”

 2) Add the 7th paragraph in Section 1 to introduce the main contribution, and as follows: 

 “Compared with other studies, the main contribution of this study is providing a new idea for the parameters optimization of vehicle braking systems. Specifically, the first is to consider both the structural and the control parameters when selecting the decision variables; the second is to set the optimal objectives from the perspective of vehicle braking performance, which is no longer limited to the brake responses. In these ways, the influences of the load transfers, the braking distribution and the parameters coupling can be decreased.”

3. The authors should be revised the conclusion with proper key results.

Response：Thanks to the reviewer’s suggestion, we have eliminated the non key results from the conclusion. The revised conclusions are as follows:

 “The results show that:

 After optimization, the step response time of maximum braking pressure is shortened by approximately 0.3 s;

 The ABS control prototype can keep the slip ratio near the ideal value at the initial braking stage. Under the action of the ABS, the optimal MFDD is improved by approximately 0.002 g;

 After the optimization, the SD of 90 km/h–0 and the Max LD are shorten by approximately 3.44 m and 0.037 m.

 In general, the EMB response time and the SD are improved obviously, and other indexes and the vehicle’s braking performance are all improved after optimization. ”

4. Figures 4, 5, 9 and 10 are not clear should be enhance.

Response: Thanks to the reviewer’s suggestion, we are also aware that the Fig 4, 5, 9 and 10 in the original manuscript are not clarity, and have improved the relevant figures again.

5. I suggest adding your main contributions comparing with other researchers clearly (what is the difference with other researchers) what is the new in your study... hope you could add it in your paper.

Response: Thanks to the reviewer’s suggestion. Actually, the main contribution and novelty of this study reflect the differences between this study and others. Therefore, in the revised manuscript, the results of previous studies are summarized, the necessity of this study are elaborated, and the comparison between this study and previous studies are carried out. The details are as follows:

 1) Analyzing the previous studies：

 “From the analyses, the optimal objectives in the [5–13] are the structural parameters, and the objectives in the [14–18] are the control parameters. Except for the [14] and the [17], the objectives in other studies are limited to improving the brake responses and reducing its size and weight.” 

 2) Elaborating the necessity of this study:

 “Actually, the performance of a brake is influenced by both the structural and the control parameters. Furthermore, the vehicle’s braking performance is not entirely determined by the brake response, but also impacted by such factors as the load transfer, the braking distribution and the control effect. Hence, the ultimate goal of the optimization should be improve the vehicle’s braking performance and not just the brake responses.”

 3) Comparing the main contributions with other studies:

 “Compared with other studies, the main contribution of this study is providing a new idea for the parameters optimization of vehicle braking systems. Specifically, the first is to consider both the structural and the control parameters when selecting the decision variables; the second is to set the optimal objectives from the perspective of vehicle braking performance, which is no longer limited to the brake responses. In these ways, the influences of the load transfers, the braking distribution and the parameters coupling can be decreased.”

6. I suggest adding future recommendations to the next researchers after the conclusion part.

Response: Thanks to the reviewer’s suggestion, we have added the future recommendations to the next researchers at the end of conclusion, and as follows:

 “Future work can address fabricating an EMB functional prototype for field and road tests to validate the optimization further. And a coordinated control and optimal distribution strategy of the EMB braking pressure maybe a meaningful study.”

---

## [Decision Letter · Decision Letter 1]

3 May 2021

Braking performance oriented multi–objective optimal design of electro–mechanical brake parameters

PONE-D-20-39894R1

Dear Dr. Li,

We’re pleased to inform you that your manuscript has been judged scientifically suitable for publication and will be formally accepted for publication once it meets all outstanding technical requirements.

Kind regards,

Jing Zhao, Ph.D.

Academic Editor

PLOS ONE

Additional Editor Comments (optional):

Reviewers' comments:

Reviewer's Responses to Questions

**Comments to the Author**

1. If the authors have adequately addressed your comments raised in a previous round of review and you feel that this manuscript is now acceptable for publication, you may indicate that here to bypass the “Comments to the Author” section, enter your conflict of interest statement in the “Confidential to Editor” section, and submit your "Accept" recommendation.

Reviewer #2: All comments have been addressed

Reviewer #3: All comments have been addressed

2. Is the manuscript technically sound, and do the data support the conclusions?

Reviewer #2: (No Response)

Reviewer #3: Yes

3. Has the statistical analysis been performed appropriately and rigorously? 

Reviewer #2: (No Response)

Reviewer #3: Yes

4. Have the authors made all data underlying the findings in their manuscript fully available?

Reviewer #2: (No Response)

Reviewer #3: Yes

5. Is the manuscript presented in an intelligible fashion and written in standard English?

Reviewer #2: (No Response)

Reviewer #3: Yes

6. Review Comments to the Author

Reviewer #2: The authors have provided satisfactory responses to all comments. I suggest that they keep the optimal slip value at a slightly lower than that corresponding to the peak \\mu value.

Reviewer #3: all comments have been addressed, the manuscript is presented in an intelligible fashion and written in standard English

7. PLOS authors have the option to publish the peer review history of their article (what does this mean?). If published, this will include your full peer review and any attached files.

Reviewer #2: No

Reviewer #3: No

---

## [Editor Report · Acceptance letter]

6 May 2021

PONE-D-20-39894R1 

Braking performance oriented multi–objective optimal design of electro–mechanical brake parameters 

Dear Dr. Li:

I'm pleased to inform you that your manuscript has been deemed suitable for publication in PLOS ONE. Congratulations! Your manuscript is now with our production department. 

Kind regards, 

on behalf of

Dr. Jing Zhao 

Academic Editor

PLOS ONE